# Translation initiation factor eIF1.2 promotes *Toxoplasma* stage conversion by regulating levels of key differentiation factors

Fengrong Wang [1], Michael J. Holmes [2,12], Hea Jin Hong[3,12], Pariyamon Thaprawat [1,4], Geetha Kannan[1], My-Hang Huynh[1], Tracey L. Schultz [1], M. Haley Licon[5], Sebastian Lourido [5,6], Wenzhao Dong[7,8,9], Jailson Brito Querido [7,8,9], William J. Sullivan Jr.[2,10], Seán E. O'Leary [3,11] & Vern B. Carruthers [1] ✉

The parasite *Toxoplasma gondii* persists in its hosts by converting from replicating tachyzoites to latent bradyzoites housed in tissue cysts. The molecular mechanisms that mediate *T. gondii* differentiation remain poorly understood. Through a mutagenesis screen, we identified translation initiation factor eIF1.2 as a critical factor for *T. gondii* differentiation. A F97L mutation in eIF1.2 or the genetic ablation of *eIF1.2* (Δ*eif1.2*) markedly impeded bradyzoite cyst formation in vitro and in vivo. We demonstrated, at single-molecule level, that the eIF1.2 F97L mutation impacts the scanning process of the ribosome preinitiation complex on a model mRNA. RNA sequencing and ribosome profiling experiments unveiled that Δ*eif1.2* parasites are defective in upregulating bradyzoite induction factors BFD1 and BFD2 during stress-induced differentiation. Forced expression of BFD1 or BFD2 significantly restored differentiation in Δ*eif1.2* parasites. Together, our findings suggest that eIF1.2 functions by regulating the translation of key differentiation factors necessary to establish chronic toxoplasmosis.

*Toxoplasma gondii* is an obligate intracellular parasite that infects felids (definitive hosts) as well as humans and other warm-blooded vertebrates (intermediate hosts)[1]. About one-third of the world's human population is seropositive for *T. gondii*[2]. Upon infection, rapidly replicating tachyzoites disseminate throughout the host. A portion of tachyzoites escape immune surveillance and differentiate into latent bradyzoite tissue cysts[3]. Bradyzoites give rise to chronic infection that can persist in the host indefinitely, possibly a lifetime[4]. Latent

bradyzoites can reconvert into proliferating tachyzoites, producing severe and potentially life-threatening tissue damage in immuno-compromised patients[2,5]. Currently, there are no effective vaccines or drugs to eliminate chronic *T. gondii* infection[6,7].

While external stresses, such as alkaline pH[8], can trigger *T. gondii* differentiation in vitro, the precise molecular mechanisms governing bradyzoite formation remain incompletely understood. The tachyzoite to bradyzoite transition entails extensive changes,

[1]Department of Microbiology and Immunology, University of Michigan Medical School, Ann Arbor, MI 48109, USA. [2]Department of Pharmacology & Toxicology, Indiana University School of Medicine, Indianapolis, IN 46202, USA. [3]Department of Biochemistry, University of California Riverside, Riverside, CA 92521, USA. [4]Medical Scientist Training Program, University of Michigan Medical School, Ann Arbor, MI 48109, USA. [5]Whitehead Institute, Cambridge, MA 02142, USA. [6]Biology Department, Massachusetts Institute of Technology, Cambridge, MA 02142, USA. [7]Department of Biological Chemistry, University of Michigan, Ann Arbor, MI 48109, USA. [8]Life Sciences Institute, University of Michigan, Ann Arbor, MI 48109, USA. [9]Center for RNA Biomedicine, University of Michigan, Ann Arbor, MI 48109, USA. [10]Department of Microbiology & Immunology, Indiana University School of Medicine, Indianapolis, IN 46202, USA. [11]Center for RNA Biology and Medicine, University of California Riverside, Riverside, CA 92521, USA. [12]These authors contributed equally: Michael J. Holmes, Hea Jin Hong. ✉e-mail: vcarruth@umich.edu

including the remodeling of the parasitophorous vacuole membrane into a highly glycosylated cyst wall[9–12], a metabolic shift from aerobic respiration to anaerobic glycolysis[13,14], the accumulation of cytoplasmic starch granules[9,15,16], and drastic alterations in gene and protein expression[17,18]. Changes in mRNA translation are required to produce key proteins that reprogram the genome for bradyzoite formation[19–22]. For instance, recent reports have demonstrated that translation of a master regulator of bradyzoite differentiation,

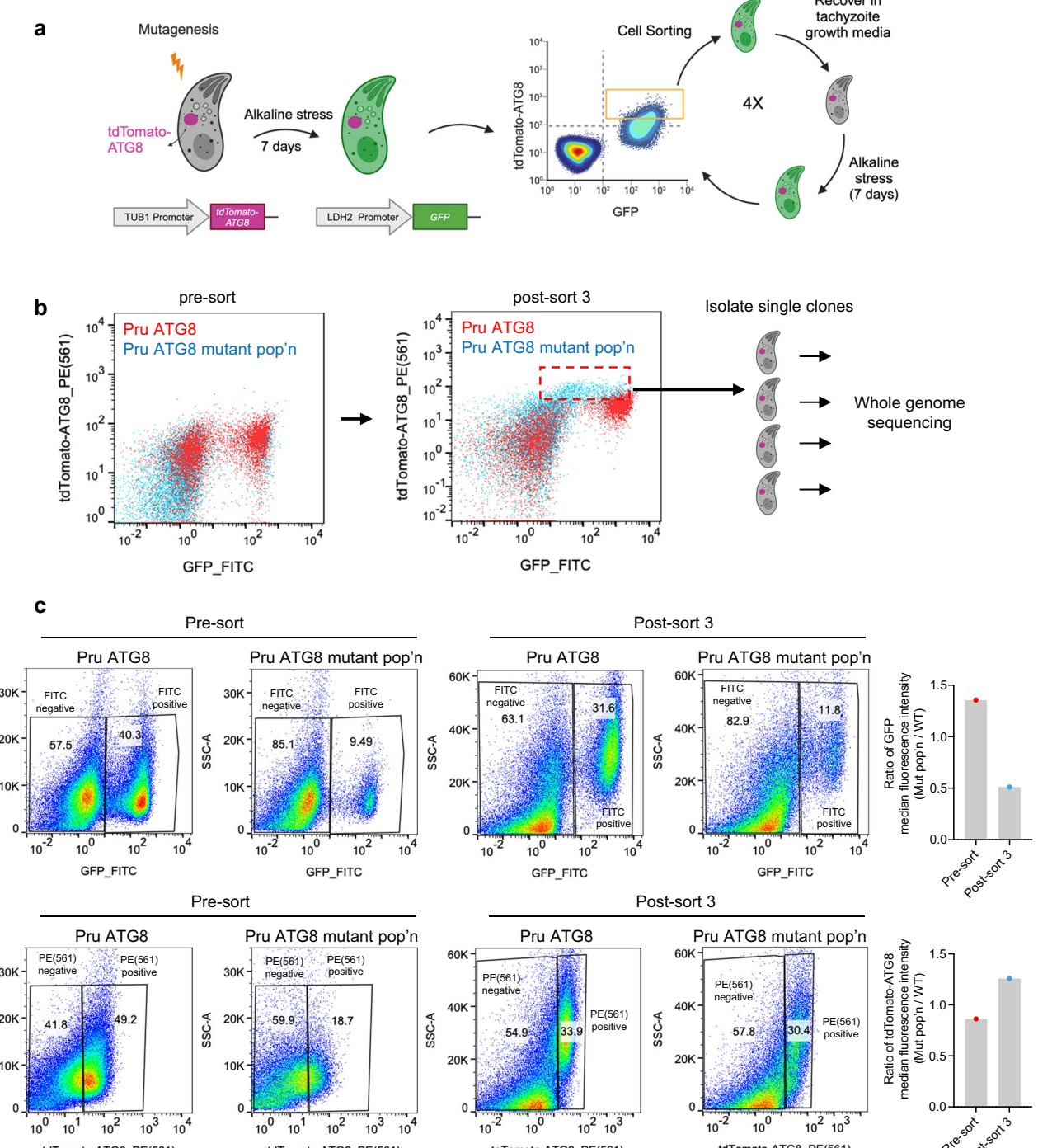

**Fig. 1 | A mutagenesis screen uncovered *T. gondii* mutants exhibiting decreased *GFP* expression driven by the *LDH2* promoter. a** Illustration of ENU-mediated mutagenesis and FACS for enriching parasite mutants with increased autophagy signal during the bradyzoites stage. **b** Representative flow cytometry pseudocolor plots demonstrating the intensities of tdTomato-ATG8 and *GFP* (driven by the bradyzoite-specific *LDH2* promoter) for parasites before or after three rounds of sorting. The final enriched population (pop'n) (red dashed box) were subcloned and subjected to whole genome sequencing. **c** Quantification of GFP and tdTomato-ATG8 intensities before and after three rounds of sorting. As different voltage settings were used for GFP and tdTomato-ATG8 for before and after the three rounds of sorting, direct comparison of absolute median fluorescence intensity values is not feasible. Therefore, median fluorescence intensities for GFP or tdTomato-ATG8 in the mutant population were divided by those in the WT to obtain the ratio of median fluorescence intensity (mut pop'n/WT), as depicted in the bar graphs. The numbers in the pseudocolor plots represent the percentage of the gated population relative to the total population. Source data are provided as a Source Data file.

BFD1[23], is regulated in part by an RNA-binding protein called BFD2[24]/ROCY1[25].

These findings add to the lines of evidence showing that translational control is critical for the differentiation of many protozoan parasites. The best-studied translation initiation factor involved in parasite differentiation is eukaryotic translation initiation factor 2α (eIF2α). Upon stress, increased eIF2α phosphorylation results in a global reduction in protein synthesis while preferentially translating specific mRNAs that manage the cell's adaptive response[26]. The phosphorylation of eIF2α has been demonstrated as crucial for stage conversions in *Leishmania*[27], *Plasmodium*[28,29], *Trypanosoma*[30], and *T. gondii*[31]. Recently, it has also been reported that the absence of eIF4E1, a key player in the cap recognition step of translation initiation, can trigger *T. gondii* differentiation[32]. With eIF4E1's function in differentiation working in parallel with that of eIF2α, the translation-directed regulatory landscape of differentiation is likely more complex than previously acknowledged.

In the current study, we discovered translation initiation factor eIF1.2 as essential for *T. gondii* cyst formation. Our findings suggest that eIF1.2 drives tachyzoite to bradyzoite stage conversion by regulating the expression of BFD1, BFD2, and possibly other factors of differentiation.

## Results

### Mutagenesis screen uncovered a mutant with a reduced ability to form bradyzoites

Previous studies reported a crucial role for autophagy in bradyzoite viability[33], and identified cathepsin protease L (CPL)[33] and ATG9[34] as pivotal genes associated with bradyzoite autophagy. Building upon these findings, the initial aim of this study was to identify additional genes involved in bradyzoite autophagy using chemical mutagenesis coupled with FACS sorting.

We performed an *N*-ethyl-*N*-nitrosourea (ENU)-mediated chemical mutagenesis screen on PruΔ*ku80LUC*tdTomato-ATG8 (Pru ATG8 hereafter) tachyzoites. This strain expresses tdTomato-ATG8 as a marker of autophagic flux and *GFP* under the control of the bradyzoite-specific LDH2 promoter[33]. After differentiating mutated tachyzoites to bradyzoites, we used fluorescence-activated cell sorting (FACS) to enrich for GFP(+) bradyzoites displaying increased tdTomato-ATG8 signal (Fig. 1a). Unexpectedly, the sorted mutant population displayed both increased tdTomato-ATG8 signal and diminished GFP signal (Fig. 1b, c). Parasites from the final enriched population were subcloned and sent for whole genome sequencing (Fig. 1b). We obtained 8 independent mutant clones with sequence coverage ranging from 48 to 74X (Supplementary Fig. 1). The mutations identified in the 8 independent clones are listed in Supplementary Data 1a–h. Notably, there were no mutations shared among these 8 independent clones. Unexpectedly, one mutant clone, 5E4, exhibited an impaired ability to form bradyzoite cysts under alkaline stress (Supplementary Fig. 2a). Clone 5E4 was validated to have significantly elevated tdTomato-ATG8 fluorescence and decreased GFP fluorescence (Supplementary Fig. 2b–m), suggesting this mutant strain has a defect in differentiation.

### F97L mutation in eIF1.2 disrupts *T. gondii* differentiation in vitro and in vivo

Mutant clone 5E4 possessed 14 unique SNVs within gene coding regions (Supplementary Data 1a). To determine if any of these mutations caused the differentiation defect, we introduced some of these SNVs individually into WT (Pru ATG8) parasites via CRISPR-Cas9 gene editing. One of the identified SNVs is F97L in a translation initiation factor (TGME49_286090) with similarity to mammalian eIF1. Non-tissue cyst-forming apicomplexan parasites have one eIF1, whereas most tissue cyst-forming apicomplexans, like *T. gondii*, possess two eIF1 paralogs (Supplementary Fig. 3a). We designated the one

recovered in our screen as eIF1.2, and the other paralog as eIF1.1 (TGME49_249370) (Supplementary Fig. 3a, b) since we reasoned that eIF1.1 might be the primary eIF1 that functions during unstressed conditions.

The F97L mutation in eIF1.2 had no significant impact on tachyzoite growth (Fig. 2a–d). This single mutation also recapitulated both the increased tdTomato-ATG8 signal as well as reduced GFP signal, indicating compromised ability to differentiate (Fig. 2e–j). Elevated tdTomato-ATG8 fluorescence was observed in both GFP(+) bradyzoites and GFP(-) tachyzoites populations, indicating that increased ATG8 signal is not linked to the differentiation defect (Supplementary Fig. 4). After 7 days of alkaline stress, eIF1.2 F97L parasites exhibited reduced formation of cyst-like structures in vitro (Fig. 2k, l). This observation is supported by their diminished ability to develop bradyzoite-specific cyst walls, as indicated by the absence of *Dolichos biflorus* agglutinin (DBA) staining (Fig. 2m, n), which recognizes the O-glycosylated cyst-wall[35]. Moreover, eIF1.2 F97L parasites showed decreased GFP expression driven by the LDH2 promoter, limited BAG1 protein expression (bradyzoite-specific), and moderately elevated SAG1 protein (tachyzoite-specific) compared to WT parasites (Fig. 2m, o–r).

We then assessed the impact of the eIF1.2 F97L mutation on acute virulence and chronic infection in C57BL/6 mice. These parasites express firefly luciferase under the control of the tachyzoite specific SAG1 promoter, allowing us to monitor WT and F97L acute stage replication in vivo by bioluminescence imaging[36]. Throughout the 15-day course of infection, tachyzoite growth remained similar, except for day 5, when mice infected with eIF1.2 F97L parasites exhibited a higher parasite burden than those infected with WT parasites (Fig. 2s). This could be due to WT parasites spontaneously differentiating under unstressed conditions, while eIF1.2 F97L mutant parasites continuing replicating as tachyzoites. Additionally, weight loss and mortality rates were comparable between mice infected with WT and mutant parasites, indicating that the F97L mutation does not impede the acute infection (Fig. 2t, u). Five weeks post-infection, mice infected with eIF1.2 F97L mutant parasites exhibited substantially lower parasite burden than those infected with WT parasites (Fig. 2v). Consistent with the in vitro analyses, these findings suggest that the F97L mutation in eIF1.2 has minimal impact on acute infection but significantly impairs establishment of chronic infection of mice.

### The eIF1.2 F97L mutation affected preinitiation complex scanning

Since eIF1.2 shares homology with mammalian translation initiation factor eIF1, we hypothesize that the F97L mutation may have perturbed translation initiation, potentially leading to a disruption in synthesis of differentiation-related proteins in stressed parasites. One of the initial steps in translation initiation is the binding of eIF1 and eIF1A to the 40 S ribosomal subunit[37] (Fig. 3a). We first performed gel shift assays to assess if eIF1.2 WT and F97L mutant can bind to the yeast 40 S ribosomal subunit. Our findings indicate no significant differences in the binding of eIF1.2 WT and F97L to the 40 S ribosomal subunit, but yeast eIF1A enhanced the binding of eIF1.2 F97L to the 40 S ribosomal subunit (Fig. 3b; Supplementary Fig. 5).

During translation initiation, the preinitiation complex (PIC) scans mRNA transcripts for the protein-synthesis start site. This is conventionally an AUG codon, but less frequently may be a near-cognate triplet such as CUG. eIF1 binds to the PIC near the decoding center of the 40 S ribosomal subunit to ensure the fidelity of start codon selection by interacting with the start codon in mRNA and anticodon in the initiator tRNA[37]. Start codon recognition triggers the rapid dissociation of eIF1 from the PIC, enabling the subsequent stages of initiation and the commencement of protein synthesis[38–40]. To investigate the effect of the eIF1.2 F97L mutation on ribosomal scanning during translation initiation, we conducted single-molecule

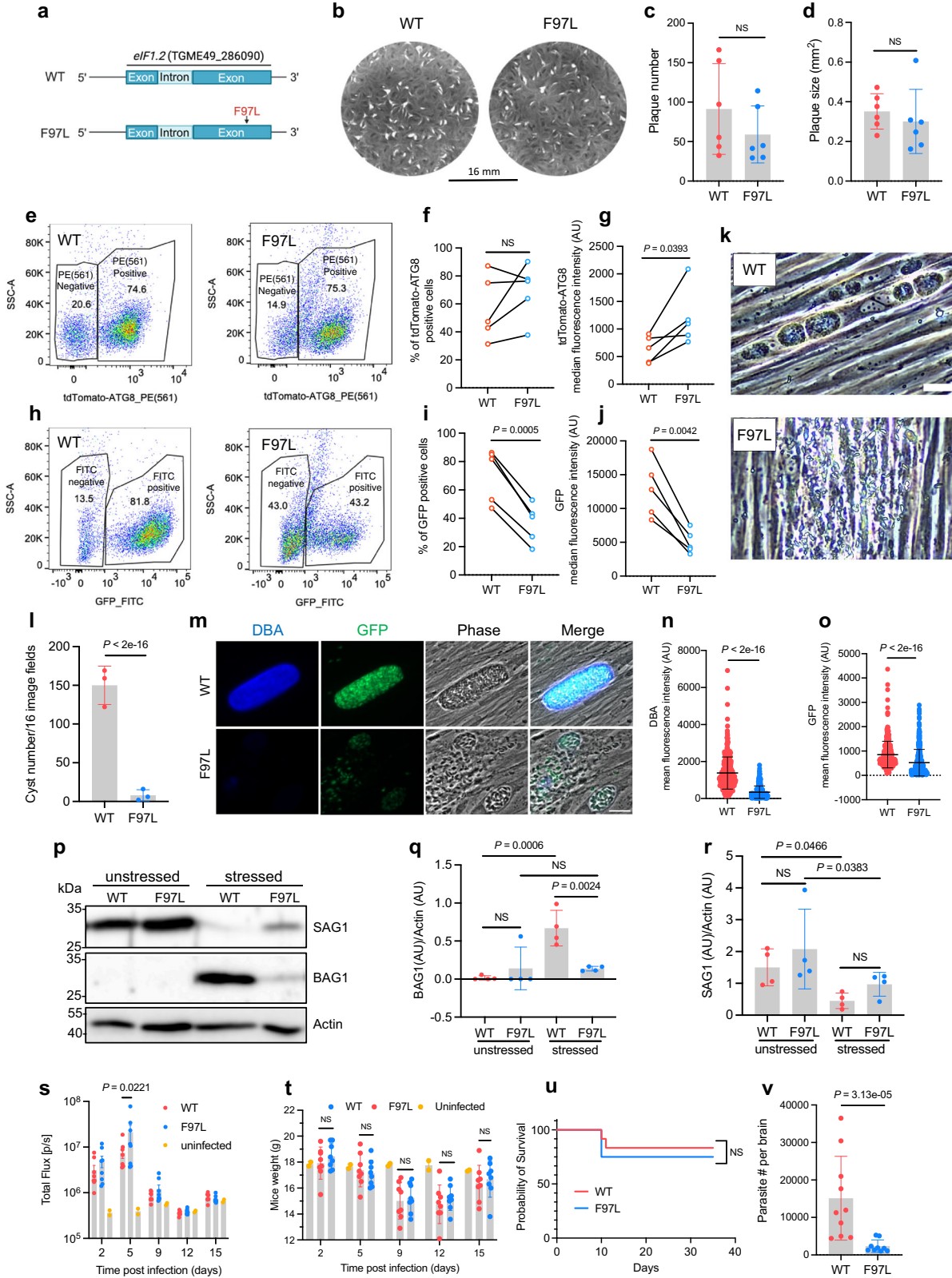

scanning assays[41] (Fig. 3c–g). In this assay, a model mRNA (Fig. 3c), fluorescently labeled with Cy5.5, is immobilized on a surface of zero-mode waveguide (ZMW) (Fig. 3d). Subsequently, a chimeric PIC, comprised of Cy5-labeled *T. gondii* eIF1.2 WT or F97L proteins, yeast 40 S ribosomal subunit labeled with Cy3, and the full complement of other yeast PIC components, is delivered to the mRNA. At the single-molecule level, the recruitment of the PIC to the mRNA leads to

simultaneous appearance of Cy3-40S and Cy5-eIF1.2 fluorescence signals. Both signals persist during PIC scanning until the start codon is recognized and the Cy5-eIF1.2 signal disappears. The time interval between co-appearance of Cy3/Cy5 fluorescence and loss of eIF1.2 fluorescence is termed the dwell time. Specifically, the dwell time quantifies the duration, following mRNA loading of the 40S−eIF1.2 complex, required for the pre-initiation complex to scan through the

**Fig. 2 | F97L mutation in eIF1.2 led to a differentiation defect in vitro and low cyst burden in vivo. a** eIF1.2 F97L mutant. **b–d** Plaque assay. Mean ± s.d. for 6 biological replicates. NS, not significant ($P > 0.05$); Student's two-tailed $t$-test. **e–o** Parasites exposed for 7 days to alkaline stress. **e–j** Flow cytometry analysis and quantification of tdTomato(+) and GFP(+) cell percentage (**f, i**) and intensity (**g, j**) in total ungated population. Numbers in (**e**) and (**h**) represent the percentage of the gated population relative to the total population. Data conducted on the same day were connected by a line. AU, arbitrary units. $n = 5$ biological replicates; NS, not significant ($P > 0.05$); Student's two-tailed $t$-test. **k** Phase contrast microscopy images. Bar, 50 μm. **l** In vitro cysts count. Mean ± s.d. for $n = 3$ biological replicates; $P$-values derived from a Poisson model. **m** Immunofluorescence images. Bar, 20 μm. **n, o** Quantification of DBA and GFP signals within individual vacuoles. Mean ± s.d. for at least $n = 300$ vacuoles from 3 biological replicates. Data were analyzed with a

linear regression model. **p–r** Western blot analysis. Stressed, 7 days of alkaline stress. Actin, loading control. Mean ± s.d for 4 biological replicates. Data were analyzed using a linear mixed model. NS, not significant ($P > 0.05$). **s** Bioluminescence imaging. Uninfected (n = 2 mice), WT ($n = 8$ mice), eIF1.2 F97L ($n = 8$ mice). Data represent mean ± s.e.m. and analyzed using a linear regression model. **t** Mice (**s**) body weight. Uninfected ($n = 2$ mice), WT ($n = 8$ mice), eIF1.2 F97L ($n = 8$ mice). Data represent mean ± s.d. and were analyzed using a linear regression model. NS, not significant ($P > 0.05$). **u** Mice survival curve. $n = 12$ mice for each group. NS, not significant ($P > 0.05$); Mantel-Cox two-sided test. **v** qPCR analysis of parasite burden in mice brain. WT ($n = 10$ mice), eIF1.2 F97L ($n = 9$ mice). Data represent mean ± s.d. and analyzed using Welch's two-tailed $t$-test. Source data are provided as a Source Data file.

5′-leader region to locate the translational initiation site, and to expel eIF1.2 (Fig. 3d, e)[41].

The mRNA used in this assay is a single-nucleotide variant of the yeast *RPL41A* mRNA, where a A to C substitution in the first position of the native start codon, 25 nucleotides from the 5′ end, results in an mRNA with a AUG at the +110 position (Fig. 3c). A short dwell time indicates recognition of the +25 CUG, while an extended dwell time suggests bypassing the CUG and recognition of downstream start codon. Consistent with gel shift data, our results demonstrate successful binding of both WT and F97L mutant *T. gondii* eIF1.2 to the yeast Cy3-40S ribosomal subunit, along with other initiation factors. This was evident from the co-arrival of the 40S-eIF1.2 complex to a single mRNA (Fig. 3f), mirroring the behavior of yeast eIF1[41]. WT eIF1.2 exhibited a higher occurrence of shorter dwell times, whereas the eIF1.2 F97L mutant displays a greater prevalence of longer dwell times (Fig. 3f, g). Our simulation data suggests that the elevated long dwell time population was not attributed to the incorporation of a very slow step (e.g., a reduced eIF1.2 ejection rate) or decelerated scanning rate (Supplementary Figs. 6, 7). Altogether, our data imply that eIF1.2 F97L does not alter the scanning kinetics but rather affects the near-cognate start-site selection.

## eIF1.2 is essential for bradyzoite formation in vitro and in vivo

To determine if the eIF1.2 F97L mutation causes a loss of function, we deleted *eif1.2* from the type II ME49Δ*ku80* strain by replacing it with a GFP expressing cassette (Δ*eif1.2*; Fig. 4a). We also complemented the Δ*eif1.2* strain by introducing an HA-tagged genomic sequence of *eif1.2* at the endogenous locus *(Δeif1.2::HA-eIF1.2*; Fig. 4a). Deletion of *eif1.2* resulted in fewer but larger plaques (Fig. 4b–d). Subsequent complementation with HA-*eif1.2* partially rescued plaque number (Fig. 4c), but not plaque size (Fig. 4d), possibly due to the influence of HA tag or prolonged passaging in cell culture. After 7 days of exposure to alkaline stress, Δ*eif1.2* parasites displayed a near-complete inability to form cyst-like structures in vitro (Fig. 4e, f), accompanied by significantly reduced DBA staining, indicating a compromised ability to form bradyzoite-specific cyst walls (Fig. 4e, g). Additionally, Δ*eif1.2* parasites exhibited a notable decrease in BAG1 expression and an increase in SAG1 levels (Fig. 4e, h, k–m). While the level of eIF1.2 protein remained unchanged after 1 day of alkaline stress, it significantly decreased after 7 days of differentiation under alkaline stress (Fig. 4i, j). Complementation with HA-*eIF1.2* restored in vitro differentiation (Fig. 4e–m). Furthermore, the complemented strain exhibited higher BAG1 levels than WT following alkaline stress, which may be attributed to the influence of HA tag (Fig. 4h, l). The impact of deleting eIF1.2 on differentiation in vitro was evident early, as WT parasites exhibited significantly higher expression levels of bradyzoite-specific genes *LDH2* and *ENO1* on day 1 post-alkaline stress compared to Δ*eif1.2* parasites. Additionally, *eif1.2* mRNA levels on day 4 were slightly but significantly higher than day 1 (Fig. 4n). To investigate the impact of eIF1.2 on cyst formation in animals, we infected CBA/J mice with WT, Δ*eif1.2*, or Δ*eif1.2::HA-eIF1.2* parasites. After 5 weeks of infection, mice

inoculated with Δ*eif1.2* parasites displayed a marked reduction in cyst numbers in brain compared to those infected with WT parasites. Complementing Δ*eif1.2* with *HA-eIF1.2* partially rescued cyst formation (Fig. 4o). Mortality rates were similar across all three strains (Fig. 4p). These results collectively suggest that eIF1.2 plays an essential role in bradyzoite formation in vitro and brain cyst burden in vivo.

## eIF1.2 governs the expression of stage-specific factors, including BFD1 and BFD2

To gain deeper insight into the role of eIF1.2 during differentiation, we first employed polysome profiling to assess the overall impact of eIF1.2 loss on protein translation in *T. gondii* after 1 day of alkaline stress. Our findings revealed well resolved polysome peaks in unstressed WT, Δ*eif1.2*, or Δ*eif1.2::HA-eIF1.2* parasites, indicating active protein translation. After 1 day of alkaline stress, all genotypes exhibited reduced polysome peaks, suggesting a downregulation of translation (Supplementary Fig. 8a, b). To scrutinize the specific changes in translation, we conducted ribosome profiling (Ribo-seq) and total RNA sequencing (RNA-seq)[42,43]. Ribosome profiling entails deep sequencing of ribosome-protected mRNA fragments (RPFs) to monitor changes in mRNA translation[44]. Translation efficiency can be calculated as the ratio of RPFs over mRNA counts mapping to each gene[42]. Principal component analyses revealed that WT and Δ*eif1.2* parasites are different from each other both in the presence and absence of stress (Supplementary Fig. 8c). Compared to their WT counterparts, both unstressed and stressed Δ*eif1.2* parasites displayed altered transcription and translation patterns (Fig. 5a, b). To globally assess whether the differentially expressed genes and proteins are associated with differentiation, we compared our Ribo-seq and RNA-seq findings with a previously published RNA-seq dataset[17] for in vivo infection (Fig. 5c–e). Among the reported 170 genes that changed at least 5 fold during the acute to chronic infection transition[17], 16 (Fig. 5d) and 37 (Fig. 5e) were also altered significantly in our dataset when comparing WT and Δ*eif1.2* parasites under unstressed and stressed conditions, respectively (Fig. 5c). A closer look at these significantly changed genes revealed that unstressed Δ*eif1.2* tachyzoites downregulated bradyzoite-specific genes, including *MAG2*, *LDH2*, *SUSA-1*, and *BAG1* at the protein translation level, and *SRS35A* at both RNA and protein translation levels (log$_2$ fold change < −1.0 and adjusted $P$ value ($P_{adj}$) < 0.05) (Fig. 5d; Supplementary Data 2a). With a few exceptions, stressed Δ*eif1.2* parasites displayed reduced transcription and translation of genes associated with chronic infection, and increased transcription and translation of genes linked to acute infection compared to their WT counterparts (Fig. 5e; Supplementary Data 2b). Similar trends were observed when comparing our Ribo-seq and RNA-seq results with a previously published RNA-seq dataset[45] for tissue cysts and in vitro tachyzoites (Supplementary Fig. 9). Overall, these findings suggest that stressed Δ*eif1.2* parasites behaves more like tachyzoites.

Stressed WT parasites upregulated *BFD1* at the protein translation level, and *BFD2* along with several bradyzoite-specific genes including *BRP1*, *LDH2*, and *ENO1* at both RNA and protein translation levels (log$_2$

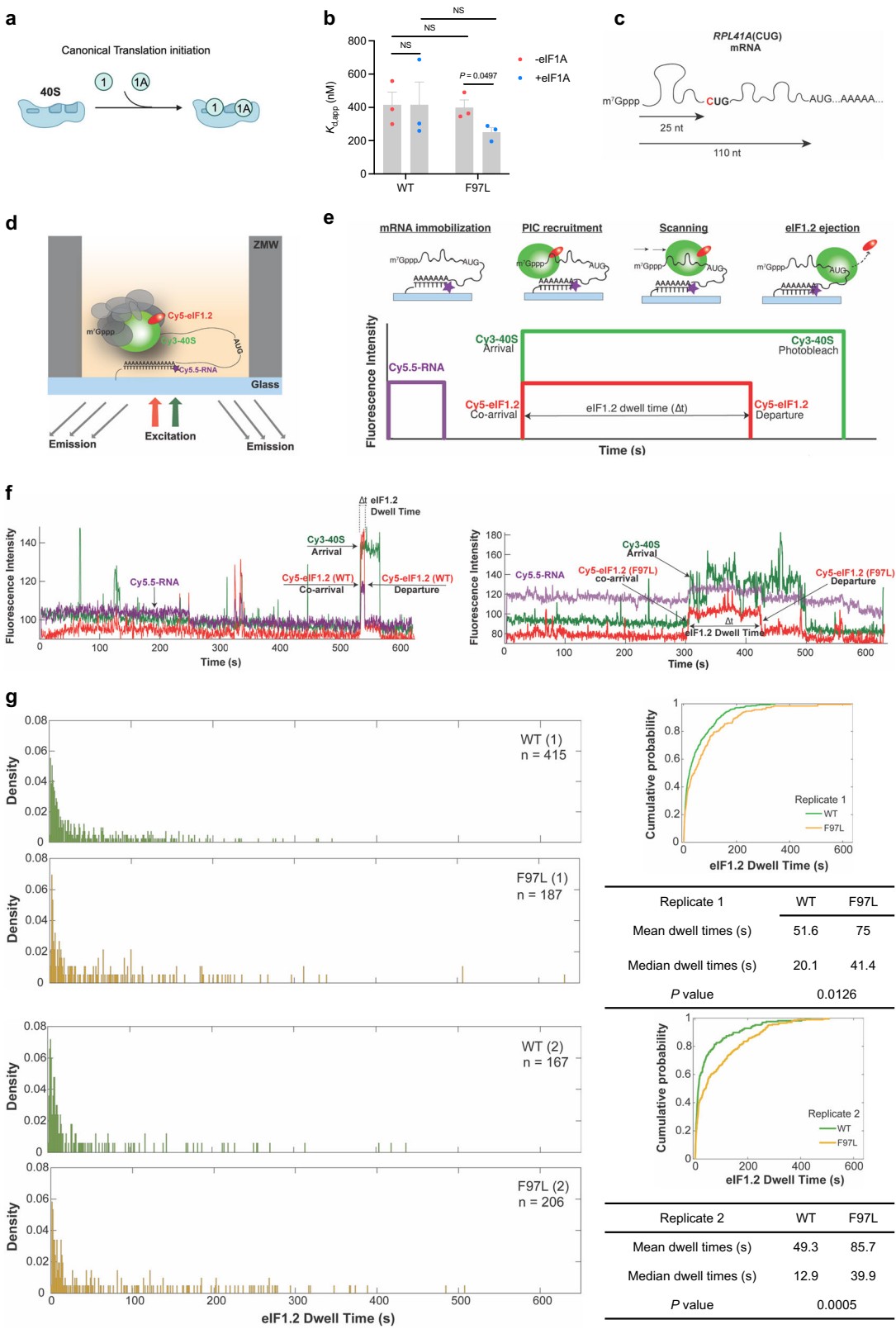

fold change > 1.0 and $P_{adj} < 0.05$). Conversely, *SAG1* was down-regulated at both RNA and protein translation levels, and the paralog *eIF1.1* was downregulated at the protein translation level in stressed WT parasites (Fig. 5f; Supplementary Data 2c). Stressed Δ*eif1.2* parasites (Fig. 5g; Supplementary Data 2d) also upregulated *BFD1* and *BRP1* at the protein translation level, as well as *LDH2* at both RNA and protein translation levels, albeit less so than WT parasites. Specifically,

whereas BFD1 translation surged by 9.3-fold in stressed WT parasites ($P_{adj} = 2.7 \times 10^{-21}$) it increased by 4.8-fold in stressed Δ*eif1.2* parasites ($P_{adj} = 7.3 \times 10^{-17}$) (Fig. 5f, g; Supplementary Data 2c, d). Notably, in response to alkaline stress, the expression of *SAG1*, *eIF1.1* and *BFD2* remained unchanged in Δ*eif1.2* parasites (Fig. 5g; Supplementary Data 2d). Furthermore, the absence of eIF1.2 had a notable impact on the translational efficiency of many mRNAs (Supplementary Fig. 8d–f,

**Fig. 3 | eIF1.2 F97L affects the scanning dynamics of preinitiation complex.**
**a** Model illustrating eIF1 and eIF1A binding to 40 S ribosome subunit during translation initiation. **b** Gel shift assay quantification. $K_{d,app}$, or the apparent equilibrium dissociation constant, measures the strength of the interaction between *T. gondii* eIF1.2 (WT or F97L) and the yeast 40 S subunit, with and without yeast eIF1A. *P*-values were calculated using Student's two-tailed *t*-test. NS, not significant ($P > 0.05$). Mean ± s.e.m for 3 biological repeats. **c** Generation of the *RPL41A*(CUG) mRNA by mutating the 5' proximal AUG site to CUG. **d** Schematic for single-molecule fluorescence experiments. Reconstituted chimeric PICs containing *T. gondii* Cy5-eIF1.2 (WT or F97L), yeast Cy3-40S, unlabeled yeast eIFs 1 A, 2, 3, 4 A, 4B, 4 G, 4E, 5, tRNA-Met, ATP, and GTP were pre-assembled and delivered to immobilized Cy5.5-*RPL41A* mRNA. ZMWs were illuminated with red and green lasers and the resulting fluorescence was collected and analyzed. **e** Cy5.5-mRNA fluorescence (purple) at the start of the movie, and terminated by a single photobleaching event, indicates a single mRNA is immobilized on the ZMW surface. mRNA recruitment of Cy5-eIF1.2/Cy3-40S PICs results in simultaneous appearance of green (40 S) and red (eIF1.2) fluorescence. The PIC then scans the mRNA until it locates an initiation site, corresponding to the time interval with sustained, continuous green and red fluorescence (denoted by an arrow). PIC mRNA start-site location triggers rapid eIF1.2 ejection, reflected by loss of red (eIF1.2) fluorescence with retention of green (40 S) fluorescence. The interval ("dwell time") between co-appearance of Cy3/Cy5 signals and loss of Cy5-eIF1.2 fluorescence is quantified across at least 100 molecules, to generate a dwell-time distribution for each experimental condition. **f** Representative traces of dwell times for WT and F97L eIF1.2. **g** Comparison of dwell time distributions, cumulative distributions, and summary (mean, median, *P*-value) for *T. gondii* WT and F97L eIF1.2 on *RPL41A*(CUG) mRNA. 2 biological replicates were performed. n in the distribution plots represents the total number of scanning events which equals to number of molecules. *P* values were calculated by Wilcoxon rank-sum two-sided test. Source data are provided as a Source Data file.

Supplementary Data 2e–g). For example, upon stress, the translational efficiency of *BFD1*, significantly increased in WT parasites by 8.4-fold ($P_{adj} = 4.2 \times 10^{-12}$; Supplementary Data 2f) and in Δ*eif1.2* parasites by 6.1-fold ($P_{adj} = 1.1 \times 10^{-12}$; Supplementary Data 2g). The translational efficiency of cyst wall protein, *CST4* (TGME49_261650)[12], remained unchanged in stressed Δ*eif1.2* parasites, but increased in stressed WT parasites by 3.15-fold ($P_{adj} = 0.0031$; Supplementary Fig. 8e, f, Supplementary Data 2f, g). Altogether, these findings imply that eIF1.2 affected expression of many stage-specific factors, including *BFD1* and *BFD2*, at the levels of transcription, translation, or both.

### The deletion of eIF1.2 impedes the translation of BFD2 induced by alkaline stress
To validate the impact of eIF1.2 on BFD1 and BFD2, we created an *eif1.2* knockout in a strain (ME49Δ*ku80*Δ*bfd1::BFD1-Ty*Δ*bfd2::HA-BFD2*)[24] in which both BFD1 and BFD2 are epitope tagged. The entire CDS of *eif1.2* was replaced with a *DHFR* selectable marker (Fig. 5h). Since BFD1 and BFD2 should exert an impact early during stage conversion, we analyzed protein levels after 1-3 days of alkaline stress. WT parasites expressed a prominent amount of BAG1 after 3 days of alkaline stress. By contrast, Δ*eif1.2* parasites failed to express BAG1 under the same conditions (Fig. 5i, l). Over the 3-day differentiation period, BFD1 and BFD2 protein levels increased significantly in WT parasites. However, the absence of eIF1.2 resulted in a modest reduction in BFD1 induction by day 3 of differentiation and a substantial hindrance to BFD2 induction in Δ*eif1.2* parasites (Fig. 5i–k).

### Conditional BFD1 or BFD2 protein expression initiates differentiation in Δ*eif1.2* parasites
To determine whether forced expression of BFD1 and BFD2 can rescue the differentiation defect of Δ*eif1.2* parasites, we conditionally expressed BFD1 and BFD2 in WT or Δ*eif1.2* parasites. We knocked out *eif1.2* in *ME49Δku80Δbfd1/HXGPRT::pTUB1-DD-BFD1-Ty*[23] (*DD-BFD1-Ty*) background. We then modified the BFD2 endogenous locus for *TUB1* promoter driven expression of *DD-HA-BFD2* in the *ME49Δku80* or *ME49Δku80Δeif1.2* background, utilizing a previously reported strategy[24] (Fig. 6a). The presence of a destabilization domain (DD domain) on the N terminus led to constitutive degradation of BFD1 or BFD2 protein until treatment with Shield-1 to stabilize the DD domain[23,24,46].

Shield-1 treatment robustly stabilized BFD1 and BFD2 proteins (Fig. 6b–d). Through western blotting quantification of whole parasite lysates, we observed significant induction of BAG1 protein by stabilizing BFD1 or BFD2 across all strains (Fig. 6b, e, f). Stabilizing BFD2 resulted in greater BAG1 induction compared to stabilizing BFD1. Loss of eIF1.2 impaired BAG1 expression induced by stabilization of BFD1 or BFD2 protein (Fig. 6b, e, f). Through fluorescence imaging of parasite vacuoles, we observed that stabilization of either BFD1 or BFD2 protein induced significant DBA and BAG1 signals in all the strains (Fig. 6g–i). DBA and BAG1 fluorescence stimulated by BFD2 stabilization considerably exceeded that caused by BFD1 stabilization. The deletion of eIF1.2 hindered DBA and BAG1 signals induced by BFD2 overexpression (Fig. 6g–i). While western blotting indicated the absence of eIF1.2 reduced BAG1 signals triggered by BFD1 overexpression, fluorescence imaging showed that BFD1 overexpression produced similar DBA and BAG1 signals in both DD-BFD1-Ty and DD-BFD1-Ty Δ*eif1.2* parasites (Fig. 6g–i). Discrepancies between western blotting and fluorescence imaging data may stem from factors such as sample size and method sensitivity. Collectively, these results suggest that stabilization of BFD1 or BFD2 can rescue differentiation in Δ*eif1.2* parasites.

## Discussion
Although the transition of *T. gondii* tachyzoites to bradyzoites is essential for long-term survival in their hosts, the precise underlying mechanism remains to be fully elucidated. In a mutagenesis screen, we identified an F97L mutation in eIF1.2 that impedes *T. gondii* differentiation in cell culture and mice. This finding is likely attributed to the fitness advantage acquired by eIF1.2 mutants, which grow akin to tachyzoites under alkaline stress conditions, compared to the parental strain that differentiates into slow-growing bradyzoites under the same conditions. Single-molecule scanning assays revealed that eIF1.2 F97L disrupted preinitiation complex scanning of a model mRNA in vitro. The observation that Δ*eif1.2* parasites also showed marked defects in differentiation in vitro and in vivo implies that the F97L mutation imparts a loss of function. RNA-seq and ribosome profiling experiments unveiled differential expression of many bradyzoite and tachyzoite-related genes in unstressed and stressed WT and Δ*eif1.2* parasites. Specifically, loss of eIF1.2 reduced BFD1 and BFD2 induction. Conditional stabilization of BFD1 or BFD2 protein triggered differentiation in Δ*eif1.2* parasites. In sum, our study revealed how the translation initiation factor eIF1.2 promotes the expression of key stage-specific factors, such as BFD1 and BFD2, facilitating *T. gondii* differentiation, as illustrated in Fig. 6j.

Non-tissue cyst forming apicomplexans possess a single eIF1. In contrast, most cyst-forming apicomplexans have two paralogs of eIF1, implying that one of these eIF1 paralogs may have evolved to regulate protein synthesis to facilitate cyst formation. Alkaline stress led to reduced translation of eIF1.1 in WT parasites, suggesting a primary function under unstressed conditions. The role of eIF1.1 needs further investigation in future studies. With altered transcription and translation profiles observed in both unstressed and stressed parasites lacking eIF1.2, eIF1.2 probably functions in both conditions. As eIF1.2 protein level remained stable in WT parasites after exposure to alkaline stress for 1 day but decreased after 7 days, future studies will be needed to unravel how eIF1.2 itself is regulated.

A single point mutation (F97L) in eIF1.2 resulted in differentiation defects like those observed in Δ*eif1.2* parasites. The F97 residue is situated in the helix a2 of eIF1.2 and is extensively conserved across various eukaryotes[40]. Mutations have been introduced in yeast eIF1 in

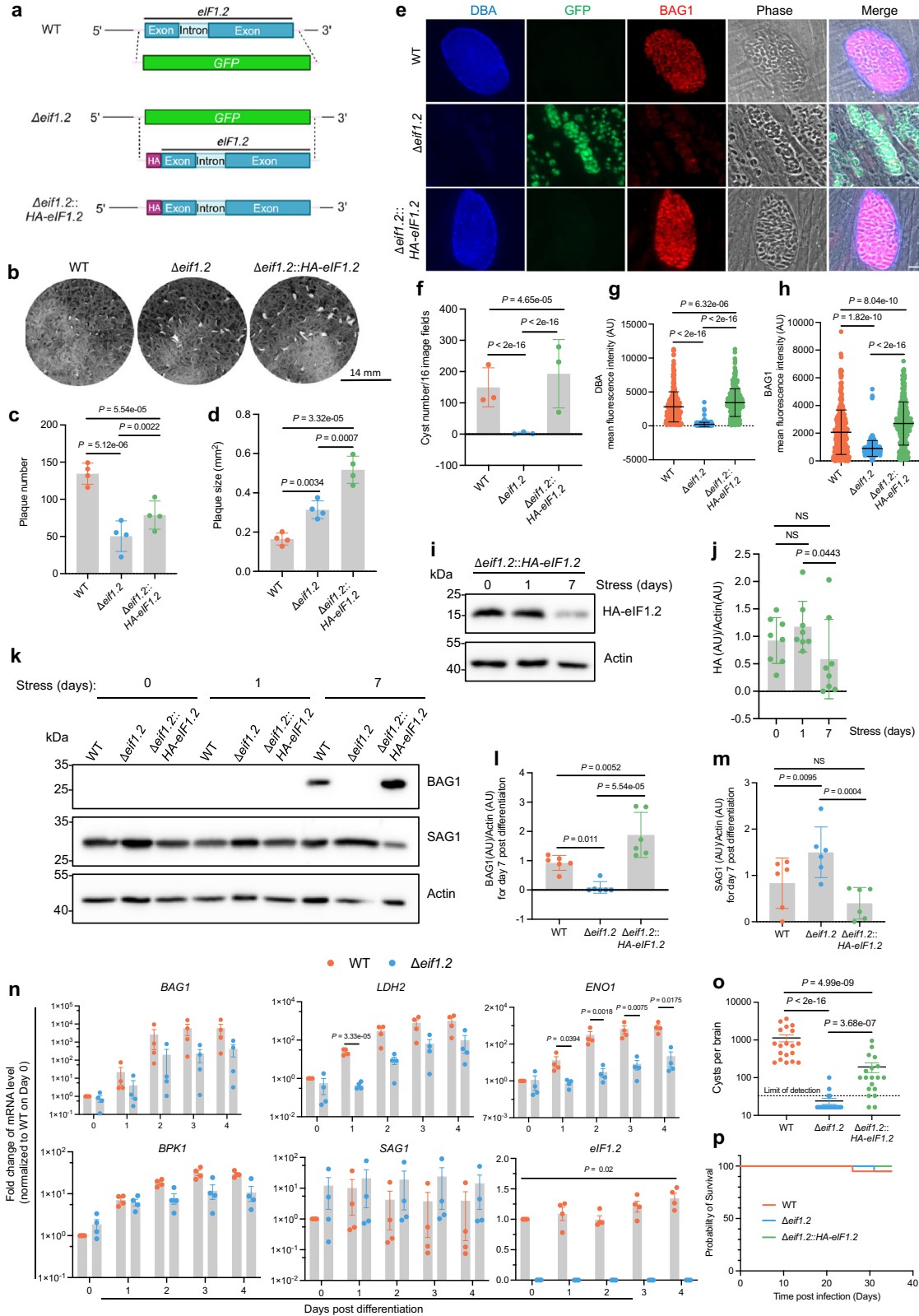

the sequence ISQLG, positioned just one amino acid downstream from this conserved phenylalanine (F). The ISQLG to ASQAA mutation reduced the binding affinity between eIF1 and the 40 S subunit, hastened the dissociation of eIF1 from the 40 S subunit, and increased the probability of initiation at a non-AUGs[47]. Our gel shift assay showed that the F97L mutation did not affect the interaction between eIF1.2 and the yeast 40 S subunit, although there was evidence that binding

was enhanced in the presence of yeast eIF1A. Single-molecule scanning assays revealed that the F97L mutation increased the propensity for the PIC to bypass a near-cognate codon CUG on the yeast *RPL41A* (CUG) mRNA. The mutation at F97 caused different effects compared to mutations at the nearby ISQLG sequence. This discrepancy may be attributed to distinct functions of these residues in the helix a2, or differences between yeast eIF1 and *T. gondii* eIF1.2. Commitment to a

**Fig. 4 | eIF1.2 is essential for tachyzoite growth and bradyzoite formation.**
**a** Generation of Δ*eif1.2* and *HA-eIF1.2* complemented strains. **b** Plaque assays.
**c**, **d** Quantification of b. Mean ± s.d. for 4 biological replicates. *P* value was calculated using a linear mixed model. **e** Representative immunofluorescence images of parasites exposed for 7 days to alkaline stress. Bar, 10 μm. **f** In vitro cysts count.
Mean ± s.d. for 3 biological replicates; A Poisson model was used to calculate *P*-value. **g**, **h** Quantification of mean fluorescence intensity for DBA and BAG1 within individual vacuoles. AU, arbitrary units. Mean ± s.d. for 3 biological replicates. A minimum of 100 vacuoles were quantified for each condition within each replicate.
Data were analyzed with a linear regression model. **i**–**m** Western blot analysis of parasites exposed for 0, 1 and 7 days to alkaline stress. Actin, loading control.
**j**, **l m** Quantification of western blot results. Data represent mean ± s.d. The HA-eIF1.2/Actin ratios were from 8 biological replicates and analyzed using a linear regression model. The BAG1/Actin or SAG1/Actin ratios were calculated from 6

biological replicates and analyzed using a linear mixed model. NS, not significant ($P > 0.05$). **n** qRT-PCR analysis of parasites exposed to alkaline stress for various days. *TUB1*, loading control. *BAG1*, *LDH2*, *ENO1* and *BPK1* are bradyzoite-specific markers. *SAG1*, tachyzoite-specific marker. Mean ± s.e.m. for 4 biological replicates.
*eIF1.2*: linear scale, other genes: logarithmic scale. *P* values for a specific day were calculated using Welch's two-tailed *t*-test combined with a Bonferroni correction.
**o** Brain cyst burden in mice at 5 weeks post-infection. The detection limit is 33 cysts per brain. Brain tissue samples with undetected cysts were assigned 16.5 cysts per brain, half the detection limit. Data represent mean ± s.e.m. Total number of mice analyzed: WT ($n = 19$), Δ*eif1.2* ($n = 20$), Δ*eif1.2::HA-eif1.2* ($n = 19$). *P*-value was calculated using a linear regression model. **p** Survival curve of infected mice. $n = 20$ mice for each group. Mantel-Cox two-sided test revealed no significant differences among the 3 infected groups. Source data are provided as a Source Data file.

particular start site also depends on its flanking nucleotide sequence, often termed the 'Kozak context'[48]. The CUG codon possesses a relatively good 'Kozak context' (CGAA). It remains unclear whether the tendency of eIF1.2 F97 to bypass this start codon is a result of increased stringency, challenges in recognizing favorable context, or other reasons. Quantitative translation initiation sequencing (QTI-seq)[49] can be conducted to unveil the global impact of the eIF1.2 F97L mutation or the loss of eIF1.2 on start codon selection.

Our studies demonstrated that parasites lacking eIF1.2 are defective in upregulating differentiation-related markers as early as 1-day post-stress, the earliest time point we examined. This implies that eIF1.2 holds a prominent role among the initial regulators for *T. gondii* differentiation. The observed influence of eIF1.2 deficiency on transcript abundances underscores the intricate interplay between translation initiation and gene expression. We propose several mechanisms through which eIF1.2 impacts mRNA levels: first, by affecting the translation of transcription factors (such as BFD1) or regulatory proteins, thus influencing the transcriptional activity of specific genes; second, by potentially triggering cellular stress responses or signaling pathways that modulate transcriptional regulation; and finally, by potentially altering the association of RNA-binding proteins with mRNA transcripts, thereby affecting mRNA stability and abundance.
Our findings suggest that eIF1.2 might play a multifaceted regulatory role in orchestrating gene expression dynamics in *T. gondii*.

Our results showed that eIF1.2 influences the translational upregulation of BFD1 and governs both transcript and translation levels of BFD2. Future experiments are needed to determine whether eIF1.2 influences BFD2 directly or indirectly through its impact on BFD1 translation, or if eIF1.2 affects a factor upstream of both BFD1 and BFD2. Furthermore, the mechanistic basis for how eIF1.2 interacts with other translational machinery and mRNA regulatory elements to induce these critical changes in protein synthesis needs investigation.

Collectively, our work identified eIF1.2 as a key regulator for *T. gondii* differentiation. Future work aimed at understanding precisely how eIF1.2 controls differentiation and how it itself is regulated will pave the way for innovative strategies to manipulate stage conversion, thereby potentially altering the course of infection and disease.

## Methods
### Ethics statement
Our research complies with all relevant ethical regulations. Experiments involving mice were reviewed and approved by the Institutional Animal Care and Use Committee at the University of Michigan, protocol number PRO00010428. Research described herein was also reviewed and approved by the Institutional Biosafety Committee at the University of Michigan, protocol number IBCA00000926.

### Parasites and host cell culture
*T. gondii* tachyzoites were cultured in human foreskin fibroblasts (HFFs, Hs27) obtained from the American Type Culture Collection

(#ATCC-CRL-1634). The culture was maintained at 37 °C under 5% $CO_2$ in D10 medium, comprising DMEM (Fisher Scientific; #10-013 cv), 10% heat-inactivated bovine calf serum (Cytiva; #SH30087.03), 2mM L-glutamine (Corning; #25-005-Cl), and 50 U/mL of Penicillin-Streptomycin (Gibco; #15070063). To induce bradyzoite formation, tachyzoites were allowed to invade HFFs overnight at 37 °C under 5% $CO_2$ in D10 medium. The next day, infected cells were then cultured at 37 °C under ambient $CO_2$ conditions in alkaline-stress medium consisting of RPMI 1640 without NaHCO$_3$ (Cytiva; #SH30011.02) supplemented with 3% heat-inactivated fetal bovine serum (Cytiva; #SH30396.03), 50 mM HEPES (Sigma; #H3375), 50 U/mL of Penicillin-Streptomycin and adjusted to pH 8.2-8.3 with NaOH. Medium was replaced daily to ensure alkaline pH[50].

### ENU-mediated chemical mutagenesis screen
On day 1, freshly cloned PruΔ*ku80LUC*tdTomato-ATG8 tachyzoites[33] were used to inoculate HFFs and were allowed to invade and replicate overnight at 37 °C under 5% $CO_2$ in standard D10 medium. On day 2, intracellular tachyzoites were mutagenized with 3.0 mM ENU (Sigma; #N3385) in D10 medium for 4 h at 37 °C. After ENU treatment, medium was removed, and the monolayer was washed 3 times with cold PBS.
Mutagenized parasites were then collected by scraping, lysing the host cells via syringing with 20 and 25-gauge needles and filtering through a 3 μm Isopore™ membrane filter (Millipore; # TSTP02500). $3.5 \times 10^5$ parasites from each sample were then added into a T175 flask containing confluent HFFs in D10 medium. Starting the following day, parasites were cultured in alkaline-stress media with ambient $CO_2$ levels to induce differentiation. Medium was replaced daily to ensure alkaline pH. At 7 days post-differentiation, bradyzoites were liberated from host cells by scrapping, syringe-lysis (20- and 25- gauge needles), pepsin treatment, filtration, and subsequent resuspension in PBS (supplemented with 2% FBS) for sorting. MoFlo Astrios cell sorter (Beckman Coulter) with Summit (v.6.3.1.16945) software was used to enrich GFP positive parasites with the top 5% tdTomato-ATG8 signal.
Sorted parasites were recovered in D10 media for a few days, followed by a 7-day exposure to exposed to alkaline-stress media before the next sorting. In total, we performed two independent mutagenesis experiments, each subjected to 4 rounds of sorting. Individual parasite clones from the 3rd and 4th sorting were obtained through serial dilution in a 96-well plate. Parasites from each clone underwent a 7-day differentiation process, followed by liberation from host cells using syringing, pepsin treatment and filtration. Subsequently, they were fixed with 4% formaldehyde for 15 min at room temperature, washed once with PBS and resuspended in PBS for analysis using an LSR Fortessa flow cytometer (BD Biosciences) with BDFACSDiVa™ (v9.0) software. Data were analyzed using FlowJo (v10.10.0) and Prism (v10.2.0) software. Genomic DNA from mutant clones with enhanced tdTomato-ATG8 signal compared to the parental strain was harvested using DNeasy Blood & Tissue Kit (Qiagen; #69506) and send for sequencing at UMICH Advanced Genomics Core. DNA quantity and

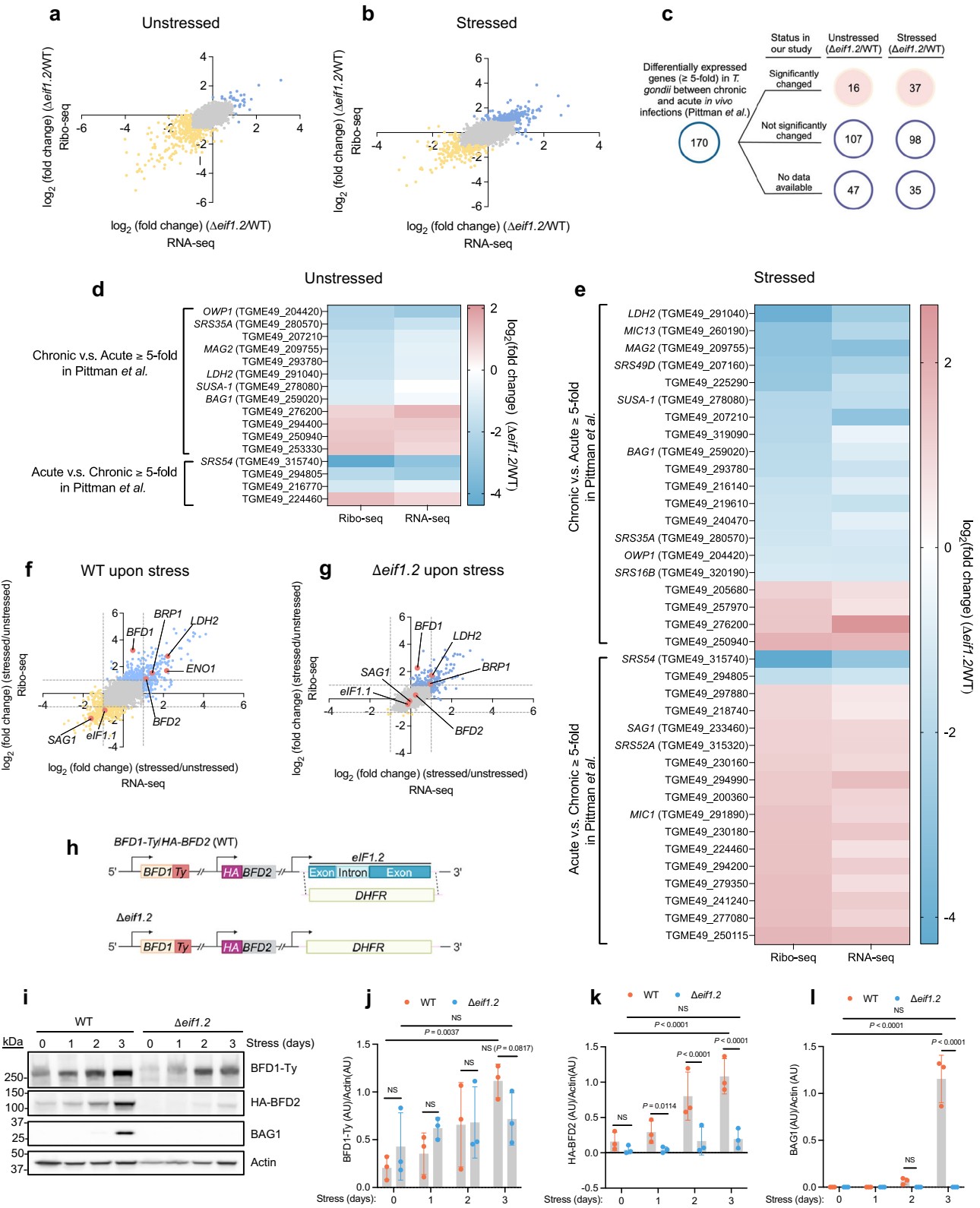

quality were assessed using Qubit™ dsDNA Quantification Assay Kits (high sensitivity, Thermo Fisher; #Q32851), and TapeStation genomic DNA reagents (Agilent; #5067-5365 and 5067-5366), respectively. Libraries were prepared using NEBNext® Ultra™ II FS DNA Library Prep Kit for Illumina (New England Biolabs; #E6177) per manufacturers protocol with 50-100 ng of DNA input and 5-6 cycles of PCR. Libraries were checked by Qubit™ dsDNA Quantification Assay Kits and TapeStation D1000 reagents (Agilent; #5067-5583), and ran on a MiSeq

to verify balance before being sequenced on the Illumina NovaSeq S4 (Paired-end, 150 bp) with BCL Convert (v4.2.4) software, according to manufacturer's recommended protocols. SNP analysis was done using the variant calling workflow (default setting) provided by VEuPathDB Galaxy Site (https://veupathdbprod.globusgenomics.org/). FastQC was used to determine the overall quality of the reads and generate reports. Based on their quality score, reads were trimmed using Sickle. The obtained reads were aligned to the *T. gondii* ME49 reference

**Fig. 5 | eIF1.2 regulates the expression of stage-specific factors, including BFD1 and BFD2. a–g** The impact of eIF1.2 deletion on transcription and translation. RNA-seq, RNA sequencing. Ribo-seq, Ribosome profiling. $n = 3$ biological replicates. **a, b, f, g** Blue dots, genes increased for at least 2-fold ($P_{adj} < 0.05$). Yellow dots, genes decreased for at least 2-fold ($P_{adj} < 0.05$). $P_{adj}$ values were derived using DeSeq2, employing a two-sided test with adjustments for multiple comparison. Grey dots, not significantly changed genes. Pink dots, genes of interest. Grey dotted lines indicate where the y-axis or x-axis equals 1 or −1. Stressed, 1 day of alkaline stress. **a** Comparison of unstressed Δ*eif1.2* and WT (ME49Δ*ku80*) parasites at transcription and translational levels. **b** Comparison of stressed Δ*eif1.2* and WT (ME49Δ*ku80*) parasites at transcription and translational levels. **c–e** Comparing our RNA-seq and Ribo-seq results with a RNA-seq dataset for in vivo acute and chronic infection[17]. Listed significantly changed genes in our study have a fold change greater than 2 or less than 0.5 in either RNA-seq, Ribo-seq or both, with $P_{adj} < 0.05$, and minimum of 5 reads. **d** Heatmap illustrates the fold change of the 16 genes that were significantly altered between unstressed Δ*eif1.2* and WT parasites in our study ($P_{adj} < 0.05$). **e** Heatmap illustrates the fold change of the 37 genes that were significantly altered between stressed Δ*eif1.2* and WT parasites in our study ($P_{adj} < 0.05$). **f** The impact of alkaline stress on transcription and translation in WT parasites. **g** The impact of alkaline stress on the transcription and translation in Δ*eif1.2* parasites. **h** Knocking out *eif1.2* in a strain[24] with both BFD1 and BFD2 tagged. **i** Western blot analysis of lysates from parasites exposed to alkaline stress for indicated days. Actin was used as a loading control. **j–l** Quantification of western blot results. Data represent the mean ± s.d. $n = 3$ biological replicates. The ratios of BFD1-Ty/Actin, HA-BFD2/Actin and BAG1/Actin were analyzed using a linear mixed model. NS, not significant ($P > 0.05$). Source data are provided as a Source Data file.

genome using Bowtie2. Coverage of the whole genome sequencing experiments was evaluated using the SAM/BAM WGS Metrics in NGS Picard (2.7.1.). Sort was used to sort alignments according to their chromosomal positions. Variants were then detected by FreeBayes, which specifically SNPs. Filter was used to refine SNP candidates, while SnpEff was utilized for variant analysis, annotate, and predicting the effects of SNPs.

## Flow cytometry assays

$1.0 \times 10^4$ to $2.5 \times 10^4$ PruΔ*ku80LUC*tdTomato-ATG8 (WT) or the 5E4 mutant clone or eIF1.2 F97L mutant were used to infect a T25 flask containing confluent HFFs in D10 medium. Starting the following day, parasites were cultured in alkaline-stress media with ambient $CO_2$ levels to induce differentiation. At 7 days post-differentiation, live parasites in T25 flasks were imaged with Olympus CKX53 microscope equipped with integrated phase contrast system using objective CACHN10x IPC/0.25 and EP50 camera. Parasites were then harvested by scraping, syringing (20- and 25-gauge needles), pepsin treatment, and filtration. Purified parasites were fixed with 4% formaldehyde for 15 min at room temperature, washed one time with PBS and resuspended in PBS for analysis on a LSR Fortessa flow cytometer (BD Biosciences) with BDFACSDiVa™ (v9.0) software. Cytometer channel used: FITC for GFP, PE (561) for tdTomato. Data were analyzed with FlowJo (v10.10.0) and Prism (v10.2.0) software.

## *T. gondii* transfection

Intracellular parasites were harvested via scraping and syringing (20- and 25- gauge needles). The parasite and host-cell debris mixture were pelleted and resuspended in cytomix buffer (2 mM EDTA, 120 mM KCl, 0.15 mM CaCl₂, 10 mM K₂HPO₄/KH₂PO₄, 25 mM HEPES, 5 mM MgCl₂, pH = 7.6). One hundred μg of guide RNA with 20 to 50 μg of repair template for each transfection were precipitated with ethanol and resuspended in cytomix buffer. The DNA was combined with the parasite and host-cell debris. Electroporation was performed in 4 mm cuvettes (Bio-Rad), utilizing GenePulser Xcell with PC and CE modules (Bio-Rad), and configured with the following parameters: 2400 V voltage, 25 μF capacitance, 50 Ω resistance.

## *T. gondii* strain generation

Oligos were synthesized either by IDT or Sigma-Aldrich. All guide RNAs were generated by substituting the original guide RNA sequence on the plasmid pCas9/sgRNA/Bleo[51] with desired guide RNA sequence, using Q5® Site-Directed Mutagenesis Kit (New England Biolabs; #E0554S). Guide RNA and repair template sequences are listed in Supplementary Data 3c.

**eIF1.2 F97L**. gRNA targeting the 2nd exon of *eIF1.2* was generated using oligos P1/P2. Repair template was generated using oligos P3/P4 to amplify a plasmid containing repair template sequence synthesized by Gene Universal. PruΔ*ku80LUC*tdTomato-ATG8[33] was transfected with 100 μg of gRNA and 20 μg of repair template. GFP⁺ parasites (gRNA plasmid express GFP-Cas9) were enriched by sorting 24–72 h post-transfection, and subcloned after recovering in D10 medium for a few days. Individual parasite clones were further characterized by PCR amplification of the locus and sequencing using P3/P4 to verify the presence of the F97L mutation.

**ME49Δku80Δeif1.2 (Δeif1.2)**. gRNA targeting the intron between the two exons of *eIF1.2* was generated using oligos P5/P6. Repair template was generated by using oligos P7/P8 to amplify GFP sequence from plasmid pCas9/sgRNA/Bleo with Phusion® High-Fidelity DNA Polymerase (New England Biolabs; #M0530S). ME49Δ*ku80*[23] was transfected with 100 μg of gRNA and 20 μg of repair template. Two days after transfection, parasites were sorted by GFP fluorescence. After recovering for a few days in D10 media, sorted parasites were subcloned. Individual parasite clones were further characterized by PCR amplification of the locus and sequencing using oligos P9/P10 to verify the deletion of *eIF1.2*.

**HA-eIF1.2**. gRNA targeting the beginning of the 1st exon of *eIF1.2* was generated using oligos P11/P12. Repair template was generated by annealing oligos P13/P14. ME49Δ*ku80* was transfected with 100 μg of gRNA and 52 μg of repair template. GFP⁺ parasites (gRNA plasmid express GFP-Cas9) were enriched by sorting 24 to 72 h post-transfection, and subcloned after recovering in D10 medium for a few days. Individual clones were screened for correct tagging using oligos P15/P16 with Phire tissue direct PCR Master kit (Thermo Fisher; #F-170S). The correct clones were further confirmed using rat anti-HA antibodies (Clone 3F10; Roche; #11867423001) in both immunofluorescence imaging and western blotting.

**ME49Δku80Δeif1.2::HA-eIF1.2 (Δeif1.2::HA-eIF1.2)**. gRNA targeting GFP was generated using oligos P17/P18. Repair template was generating by using oligos P19/P20 to amplify genomic sequence of HA-eIF1.2 strain. One hundred μg of gRNA and 43 μg of repair template were transfected into ME49Δ*ku80*Δ*eif1.2* parasites. The bleomycin resistant cassette on the gRNA allowed for the selection of transfected parasites. One day after transfection, parasites were treated with 50 μg/ml of phleomycin for 4 h. Following growth in D10 medium for a few days, selected parasites were subcloned. Phire tissue direct PCR Master kit with oligos P21/P10 were used to screen individual clones for the presence of GFP. GFP⁻ clones were further confirmed by PCR amplification of the locus and sequencing using oligos P9/P10 to verify the absence of GFP.

**ME49Δku80Δeif1.2Δbfd1::BFD1-TyΔbfd2::HA-BFD2 (BFD1-Ty/HA-BFD2Δeif1.2)**. gRNA targeting the beginning of the 2nd exon of *eIF1.2* was generated using oligos P22/P23. Repair template was generated by using oligos P24/P25 to amplify *DHFR* cassette from pDHFR-TSc3/M2M3[52]. One hundred μg of gRNA and 76 μg of repair template were transfected into ME49Δ*ku80*Δ*bfd1*::BFD1-TyΔ*bfd2*::HA-BFD2[24]. One

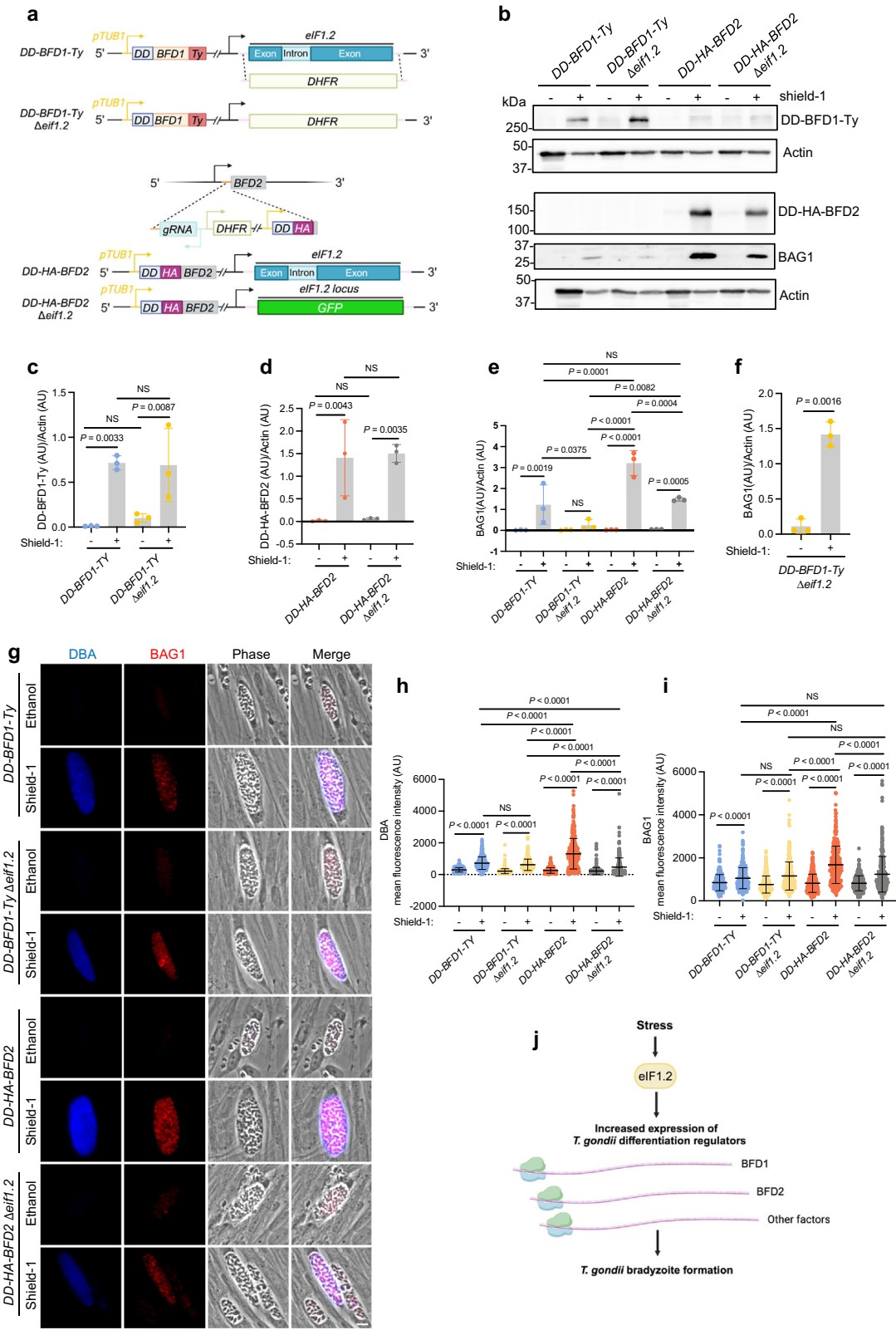

day after transfection, parasites underwent a week-long selection with 3 μM of pyrimethamine before subcloning. Phire tissue direct PCR Master kit with oligos P9/P26 were used to screen single clones for the presence of *DHFR*.

**ME49Δku80Δeif1.2Δbfd1/HXGPRT::pTUB1-DD-BFD1-Ty (DD-BFD1-TyΔeif1.2).** The same gRNA, repair template and selection method

employed for generating ME49Δ*ku80*Δ*bfd1::BFD1-Ty*Δ*bfd2::HA-BFD2*Δ*eif1.2* were also used for deleting *eIF1.2* in *ME49Δku80Δbfd1/HXGPRT::pTUB1-DD-BFD1-Ty*[23].

**ME49Δku80 DD-HA-BFD2 (DD-HA-BFD2) and ME49Δku80Δeif1.2 DD-HA-BFD2 (DD-HA-BFD2Δeif1.2).** BFD2 was conditionally overexpressed in ME49Δ*ku80* and ME49Δ*ku80*Δ*eif1.2* strains following a

**Fig. 6 | Stabilization of BFD1 or BFD2 triggers differentiation in Δ*eif1.2* parasites. a** Stabilization of BFD1 or BFD2 in WT or Δ*eif1.2* parasites. **b** Western blot analysis of lysates from parasites treated with vehicle control (100% ethanol) or 3 μM Shield−1 for 4 days. Actin, loading control. The actin and DD-BFD1-Ty were from the same sample but run on separate blots due to a bubble in one of the original actin bands on the same blot as DD-BFD1-Ty. Uncropped images for both the current actin and previous actin blots are provided in the Source Data for (**b**). **c**−**f** Quantification of western blot results. AU, arbitrary units. Data represent the mean ± s.d. *n* = 3 biological replicates. The DD-BFD1-Ty/Actin ratio was analyzed using a linear regression model. The DD-HA-BFD2/Actin or BAG1/Actin ratios were analyzed using a linear mixed model. NS, not significant (*P* > 0.05). **e** The BAG1 signals were captured when BAG1 bands in all lanes in the western blots were not saturated. Here, BAG1 levels in Shield−1 treated *DD-BFD1-Ty*Δ*eif1.2* were notably

low and showed no significant difference compared to the vehicle-treated group. **f** With longer exposure of the western blots to ensure BAG1 signals for *DD-BFD1-Ty*Δ*eif1.2* parasites not saturated, Shield−1 treated *DD-BFD1-Ty*Δ*eif1.2* exhibited significantly higher BAG1 levels compared to the vehicle treated control. **g** Representative vacuoles of parasites treated with vehicle or 3 μM Shield−1 for 4 days. Bar, 10 μm. **h, i** Measurement of mean fluorescence intensity for DBA and BAG1 within individual vacuoles. Mean ± s.d. plotted for 3 biological replicates. A minimum of 100 vacuoles were quantified for each condition within each replicate. Data were analyzed with a linear mixed model. **j** Model for the role of eIF1.2 in *T. gondii* differentiation. Upon stress, eIF1.2 enhances the expression of key bradyzoite-specific factors, such as BFD1 and BFD2, to drive *T. gondii* differentiation. Source data are provided as a Source Data file.

previously described method[24]. Briefly, targeted integration of a linearized construct was achieved using a gRNA directed at the 5' end of the BFD2 coding sequence. The integration at BFD2 endogenous locus allowed for tagging BFD2 with the Shield-1-stabilized DD domain and HA, as well as the substitution of the native promoter with the α-tubulin (pTUB1) promoter.

## Plaque assays

Either 200 tachyzoites from the Pru background or 500 tachyzoites from the ME49 background were added to individual wells in 6-well plates containing confluent HFFs in D10 medium. Parasites were allowed to grow undisturbed for 10 days. Plaque numbers were counted under Olympus CKX53 microscopy equipped with integrated phase contrast system using Olympus plan N 2x/0.06 Objective. For Pru strain, the plaque number in each biological replicate was calculated as the mean of counts from three wells in 6-well plates. For ME49 strain, the plaque number in each biological replicate was calculated as the mean of counts from four wells in 6-well plates. Plates were then fixed and stained with crystal violet solution (0.2% of crystal violet and 70% of EtOH) for 10 min at room temperature, rinsed with $H_2O$ and imaged using Molecular Imager® Gel Doc™ XR System (Bio-Rad) with Image Lab™ (v6.0.1) software. Plaques were manually outlined and measured for size using ImageJ. For pru strain, the plaque size for each biological replicate was determined as the mean of average plaque sizes from three wells in 6-well plates. In total, the plaque sizes of *n* = 1281 (WT) and *n* = 759 (F97L mutant) were quantified for the 6 biological replicates. For ME49 strain, the plaque size for each biological replicate was determined as the mean of average plaque sizes from four wells in 6-well plates. In total, the plaque sizes of WT (*n* = 535), Δ*eif1.2* (*n* = 448) and Δ*eif1.2*::*HA-elF1.2* (*n* = 812) were quantified for the 4 biological replicates.

## Immunoblotting

Unstressed parasites were cultured in HFFs in D10 medium (5% $CO_2$) for 48 h. Alkaline-stressed parasites were initially grown in HFFs in D10 medium (5% $CO_2$) for 24 h and then exposed to alkaline-stress medium (ambient $CO_2$) for varying days. Unstressed PruΔ*ku80*-*LUC*tdTomato-ATG8 WT and F97L mutant parasites were liberated from host cells via scraping, syringing (20- and 25-gauge needles), and filtration. 7-day stressed PruΔ*ku80LUC*tdTomato-ATG8 WT and F97L mutant parasites were harvested via scraping, syringing (20- and 25-gauge needles), pepsin treatment, and filtration. Unstressed or 1-day stressed ME49Δ*ku80*, Δ*eif1.2*, Δ*eif1.2*::*HA-elF1.2*, as well as unstressed or stressed DD-BFD1-Ty/DD-HA-BFD2 and DD-BFD1-Ty/DD-HA-BFD2Δ*eif1.2* parasites, were harvested via scraping and syringing (20- and 25-gauge needles). 7-day stressed ME49Δ*ku80*, Δ*eif1.2*, Δ*eif1.2*::*HA-elF1.2* parasites were harvested via scraping, syringing (20- and 25-gauge needles), pepsin treatment, and filtration. DD-BFD1-Ty, DD-BFD1-TyΔ*eif1.2*, DD-HA-BFD2, DD-HA-BFD2Δ*eif1.2* parasites were inoculated into T25 flasks with confluent

HFFs in D10 medium (5% $CO_2$). After 4 h of invasion, the medium was replaced with fresh D10 medium containing either the vehicle control (100% ethanol) or 3 μM Shield-1. These parasites were cultured for additional 4 days and then harvested by scraping and syringing (20- and 25-gauge needles).

Parasites were lysed with RIPA buffer (Thermos Scientific; #89900) supplemented with protease inhibitor cocktail (Roche; #11836153001) for 10 min on ice with rocking. Lysates were centrifuged at 4 °C for 10 min at maximum speed (21,130 x g). Supernatant from each sample was transferred to a new tube. Each supernatant was supplemented with 5X SDS-PAGE sample buffer and 10% β-mercaptoethanol, resulting in a final concentration of 1X SDS-PAGE buffer and 2% β-mercaptoethanol. Each sample mixture was boiled at >90 °C for 5 to 10 min and then stored at −20 °C until use.

Protein lysates from both unstressed and stressed parasites, including PruΔ*ku80LUC*tdTomato-ATG8 WT and F97L mutant, ME49Δ*ku80*, Δ*eif1.2*, Δ*eif1.2*::*HA-elF1.2*, were subjected to electrophoresis using 15% polyacrylamide SDS-PAGE gels and transferred onto 0.45-μm nitrocellulose membranes (Bio-Rad) with Trans-Blot® SD semi-dry transfer cell (Bio-Rad) for 30 min at 16 V at room temperature. Protein lysates from both unstressed and stressed parasites, comprising DD-BFD1-Ty/BB-HA-BFD2Δ*eif1.2*, as well as from vehicle or Shield-1 treated parasites, which included DD-BFD1-Ty, DD-BFD1-TyΔ*eif1.2*, DD-HA-BFD2, DD-HA-BFD2Δ*eif1.2* were subjected to electrophoresis using 10% polyacrylamide SDS-PAGE gels and transferred onto 0.45 μm nitrocellulose membranes (Bio-Rad) with wet/tank transfer system (Bio-Rad) for 1 h and 20 min at 100 V at 4 °C.

Following transfer, membranes were blocked with 5% milk in PBS (with 0.05% Triton X-114 and 0.05% Tween-20) for 30 min at room temperature. Primary antibodies were diluted in 1% milk in PBS (with 0.05% Triton X-114 and 0.05% Tween-20) and applied to membranes overnight at 4 °C. Primary antibodies used include rabbit anti-BAG1[34] (1:5000; Carruthers Lab), rat anti-SAG1[34] (1:5000; Caruthers Lab), rabbit anti-TgActin[53] (1:20,000; Sibley lab, Washington University in St. Louis), mouse anti-TgActin[54,55] (1:2000; Soldati Lab, University of Geneva), rat anti-HA (Clone 3F10; 1:2500; Roche; #11867423001), mouse anti-Ty[56] (Clone BB2; 1:10,000, Sibley lab, Washington University in St. Louis). Membranes were washed 3 times with PBS (0.05% Tween-20) before incubation with HRP-conjugated secondary antibodies (1:5000) for 1 h at RT. Proteins were detected using SuperSignal™ West Pico PLUS Chemiluminescent Substrate or Femto Maximum Sensitivity Substrate (Thermo Fisher; #1863096 or #34095). The Syngene PXi6 imaging system with Genesys (v1.8.2.0) software was used to detect signals, and Fiji[57] (v2.9.0/1.53t) software was used for quantification.

## Mouse experiments

Mice were housed in rooms with a 12 h light/dark cycle, with lights on from 5 am (Daylight Saving Time) / 6 am (Eastern Standard Time) to 5 pm (Daylight Saving Time) / 6 pm (Eastern Standard Time). The

ambient temperature was maintained at 22.2 °C, and humidity levels ranged between 30 and 70%.

**Bioluminescence imaging.** Six- to eight-week-old albino C57BL/6 female mice (Jackson Laboratories) were infected intraperitoneally with $2 \times 10^5$ PruΔ*ku80LUC*tdTomato-ATG8 (WT) and F97L mutant parasites diluted in PBS. Bioluminescence imaging was performed on days 2, 5, 9, 12, and 15 following infections using the IVIS® Spectrum in vivo imaging system (PerkinElmer) with Living Image® (v4.0) software. Mice were intraperitoneally injected with 200 µl of 40 mg/ml D-luciferin (Promega) in PBS and anesthetized with isoflurane. Imaging was initiated 10 min after the administration of D-luciferin. Mice were imaged ventrally. Bioluminescence signal intensities were quantified using the Living Image® software. Body weight was measured daily after imaging. $n = 2$ for uninfected mice, $n = 8$ for mice infected with WT parasites, $n = 8$ for mice infected with mutant parasites.

**Quantification of brain parasite burden by qPCR.** Six- to eight-week-old C57BL/6 female mice (Jackson Laboratories) were infected intraperitoneally with $2 \times 10^5$ PruΔ*ku80LUC*tdTomato-ATG8 (WT) and F97L mutant parasites diluted in PBS. After 5 weeks of infection, brains were minced with scissors, vortexed, and homogenized in TRizol™ Reagent (Invitrogen, #15596018) by syringing the mixture through a 20-gauge needle. Genomic DNA was isolated according to the manufacture's protocol for DNA extraction, and then quantified using a NanoDrop spectrophotometer. qPCR was performed using 100 ng to 300 ng of DNA in SsoAdvanced™ Universal SYBR® Green Supermix (Bio-Rad; #172-5271) and primers for the multicopy B1 gene of *T. gondii*[58]. The PCR reactions were performed with CFX96 Touch Real-Time PCR Detection System (Bio-Rad) with CFX Manager (v3.1) software using the following parameters: 3 min at 98 °C, and 40 cycles of 15 s at 98 °C, 30 s at 58.5 °C, and 30 s at 72 °C. A standard curve was built with $10^0$, $10^1$, $10^2$, $10^3$, $10^4$, and $10^5$ tachyzoites. The parasite count was determined by comparing the average Cq value of the standards with that of the tested samples. Total number of mice used for infection to generate the survival curve is as follows: WT ($n = 12$), eIF1.2 F97L ($n = 12$). As illustrated in the survival curve (Fig. 2u), two mice infected with WT parasites and three mice infected with eIF1.2 F97L parasites died during the 5-week infection period. The total number of mice that survived until last day of the 5-week infection and were used to analyze brain parasite burden were as follows: WT ($n = 10$), eIF1.2 F97L ($n = 9$).

**Counting mouse brain cysts.** Seven- to eight-week-old old CBA/J female mice (Jackson Laboratories) were randomized and infected intraperitoneally with 500 tachyzoites of ME49Δ*ku80*, ME49Δ*ku80*Δ*eif1.2*, or ME49Δ*ku80*Δ*eif1.2::HA-eIF1.2* strains diluted in PBS.

Data presented in Fig. 4o were pooled from four biological replicates. At 5 weeks post-infection, mice brains were minced with scissors, vortexed and homogenized in 1 mL of ice-cold PBS by syringing the mixture through a 20-gauge needle. To ensure unbiased analysis, mice were coded, obscuring the identity of each sample. Three counts of 10 µl each, totaling 30 µl (equivalent 3% of the brain), from the brain homogenate of each infected mouse, were analyzed using phase-contrast microscopy for cyst enumeration. The total cysts per mouse brain were then calculated. For samples in which no cysts were found in the 30 µl of homogenate, a value of 16.5 cysts per brain was assigned, which corresponds to half of limit of the detection (33 cysts per brain). Total number of mice used for infection to generate the survival curve is as follows: WT ($n = 20$), Δ*eif1.2* ($n = 20$), and Δ*eif1.2::HA-eIF1.2* ($n = 20$). As illustrated in the survival curve (Fig. 4p), one mouse infected with WT parasites and one mouse infected with Δ*eif1.2::HA-eIF1.2* parasites died during the 5-week infection period. The total number of mice that survived until

last day of the 5-week infection and were used to analyze brain cysts were as follows: WT ($n = 19$), Δ*eif1.2* ($n = 20$), and Δ*eif1.2::HA-eIF1.2* ($n = 19$).

**Phylogenetic analysis of eIF1**
Protein sequences from representative apicomplexan parasites were aligned using ClustalW, and the phylogenetic tree was generated using the neighbor-joining method with 1000 bootstrap replications in MEGA11. The visualization of the tree was generated by uploading the tree from MEGA11 (v11.0.13) into ITOL (v6). The visualization of proteins sequences alignment was generated using ClustalW in MegAlign Pro (v17.5.0, DNASTAR).

**eIF1.2 WT and F97L purification**
*T. gondii* eIF1.2 purification was adapted from the previously established protocol[59]. *T. gondii* WT and F97L eIF1.2 mutant coding sequences were separately inserted into a pTYB2 vector, between the NdeI and XmaI restriction sites, to produce a sequence encoding eIF1.2 with a C-terminal fusion to a self-cleavable intein and a chitin-binding domain. The plasmids were transformed into BL21-codon plus RIL cells (Agilent; #230245) and were plated onto an LB agar plate with ampicillin (100 µg/mL). A single colony of the transformed cells was grown overnight in 10 mL of LB with ampicillin (100 µg/mL) at 37 °C. The culture was then used to inoculate 1 L of LB with ampicillin (100 µg/mL) at 37 °C and was grown to OD600 of 0.6 by shaking at 180 rpm. The culture was induced with 0.3 mM IPTG and was grown overnight at 16 °C. The cells were spun down (2831 x g for 15 min) and were stored at −80 °C until the day of purification. For protein purification, the cells were resuspended in 30 mL of intein lysis buffer (20 mM HEPES-KOH (pH 7.4), 0.5 M KCl (pH 7.6), 0.1 % TritonX-100, 1 mM EDTA, protease inhibitors) and were lysed using a sonicator (Branson Sonifier 450; 6 cycles of 30 s at 90 % amplitude and setting 5, with 5 mm tip and 2 min rest between each cycle). The lysate was clarified by centrifugation at 39191 x g for 30 min at 4 °C. The supernatant was filtered with a 0.22 µm PVDF syringe filter (Corning; #975413) and was loaded onto 2 mL of chitin resin (NEB; #S6651S) pre-equilibrated with lysis buffer. The clarified lysate and resin mixture was incubated for an hour while rocking at 4 °C. The resin was transferred onto a polypropylene column and was washed with 15 mL of lysis buffer followed by 15 mL of intein wash buffer (20 mM HEPES-KOH (pH 7.4), 1 M KCl (pH 7.6), 0.1 % TritonX-100, 1 mM EDTA) and 15 mL of intein cleavage buffer without MESNA (20 mM HEPES-KOH (pH 8), 0.5 M KCl, 1 mM EDTA). The protein-bound resin was incubated with 0.5 mL of intein cleavage buffer containing 200 mM MESNA and dipeptide-fluorophore [Cys-Lys(ε-Cy5)] overnight at room temperature. The protein was eluted the next day and was buffer exchanged into a low salt buffer (75 mM HEPES, pH 7.6, 100 mM KCl, 10% glycerol, 2 mM DTT) using a 10-DG column (Bio-rad) for further purification. The protein was loaded on a 1 mL Heparin-HP column (Cytiva) pre-equilibrated with low-salt buffer and was eluted with a gradient (0–100%) of high-salt buffer (75 mM HEPES, pH 7.6, 1 M KCl, 10% glycerol, 2 mM DTT). The eluted samples were analyzed using 15 % SDS-PAGE and the pure fractions were combined. The samples were buffer exchanged into storage buffer (20 mM HEPES-KOH pH 7.4, 100 mM KOAc, 10 % Glycerol, 2 mM DTT) using a 5-KD concentrator (Sartorius; #4558705). The purified samples were aliquoted, flash-frozen with liquid N2, and stored at −80 °C.

**Electrophoretic mobility shift assay**
To quantify the relative binding of *T. gondii* eIF1.2 WT and F97L to the *Saccharomyces cerevisiae* 40 S ribosomal subunit, a native-PAGE gel shift assay was performed as described previously[41,59]. Cy5-eIF1.2 WT or F97L (12.5 to 800 nM; 12.5 to 400 nM for + eIF1A condition) was added to purified 40S subunits (100 nM) in 1× binding buffer (34 mM Tris, 57 mM HEPES-KOH, 0.1 mM EDTA, 2.5 mM MgCl2, 2 mM DTT, pH 7.5, 10 % glycerol, 100 mM KOAc) in a final volume of 15 µL. For assays with

*S. cerevisiae* eIF1A, the protein was included at 1 µM in each binding reaction. Reactions were incubated at room temperature for 30 min, then loaded onto a 4% native PAGE gel (37.5:1 acrylamide:bis-acrylamide) prepared with 1× binding buffer. The gel was electrophoresed at 80 V for 1.5 h at 4 °C, then eIF1.2 fluorescence was visualized using a Typhoon imager (Cytiva) equipped with a Cy5 fluorescence filter, using ImageQuantTL (v10.2.499) software. 40S–eIF1 complex band intensity was measured using Fiji[57] software. A binding curve was generated by fitting the measured 40 S•eIF1.2 (or 40 S•eIF1A•eIF1.2) intensities, *I*, to the Langmuir isotherm:

$$I = \frac{[\text{eIF1.2}]}{K_{\text{d,app}} + [\text{eIF1.2}]} \tag{1}$$

where *I* is the background-corrected fluorescence intensity, [eIF1.2] is the eIF1.2 concentration, and $K_{\text{d,app}}$ is the apparent equilibrium dissociation constant. Although the 40S subunit concentration (100 nM) was higher than the apparent dissociation constant, goodness-of-fit analysis indicated that a quadratic binding isotherm did not reliably describe the titration data.

### Single-molecule scanning assays

Prior to scanning assays, full-length yeast mRNA (*RPL41A*) was prepared by in vitro T7 runoff transcription, 5′-m7Gppp-capped, and 3′-polyadenylated. The transcript included an appended 5′-terminal GpG dinucleotide at the transcript +1 and +2 positions, owing to use of the T7 promoter in the transcription construct. The poly(A)-tailed RNA was then hybridized to a 100-mer oligo(dT) conjugated to biotin and Cy5.5 fluorophore at its 5′ and 3′ ends, respectively. To investigate the impact of a near-cognate start codon on scanning, the 5′ proximal AUG start site on the mRNA was mutated to CUG. Single-molecule scanning experiments were conducted in zero-mode waveguide (ZMW) arrays housed on Pacific Biosciences SMRT cells and imaged with a customized Pacific Biosciences RS II instrument equipped with Proprietary Pacbio software[60], available at the University of California, Riverside. The procedure was performed as reported previously[41]. It is noteworthy that the current PacBio Sequell II and Revio systems lack this capability, making the RS II system the sole option for conducting these experiments. Briefly, the mRNA was immobilized on the NeutrAvidin-coated surface of the ZMW via the annealed biotinylated oligo(dT). SMRT cells with immobilized mRNAs were illuminated 532 nm and 642 nm lasers, to excite Cy3 and Cy5, respectively. The reconstituted chimeric PIC composed of *T. gondii* Cy5-eIF1.2 (WT or F97L separately for testing each condition), and yeast Cy3-40S ribosomal subunit, eukaryotic initiation factors 1 A, 3, 5, 4 A, 4B, 4 G, ATP, and the ternary complex (eIF2, GTP, and tRNA) was robotically delivered to the immobilized mRNA by the instrument. Movies were recorded for 15 min at 10 frames per second. Single-molecule fluorescence traces were extracted from the raw movie data by previously-described custom MATLAB scripts[60]. Traces containing single mRNA molecules were first manually curated by visual confirmation that a Cy5.5-mRNA signal at the start of the movie was followed by a single Cy5.5 photobleaching event. Within this set, traces containing PIC–mRNA recruitment events of interest were further selected by visual confirmation of the presence of events where *T. gondii* Cy5-eIF1.2 co-arrived within the same movie frame as Cy3-40S. The duration of the Cy5-eIF1.2 signal within these co-arrival events was then manually assigned. eIF1.2 dwell-time histograms representing the normalized probability density binned at 1 second, and cumulative probability distributions, were both created using MATLAB (v2017a)[61].

### Simulation model

Wolfram Mathematica (v13.1) software was utilized to run the simulations. Scanning was modeled as a process consisting of *N* sequential irreversible steps, with each step reflecting forward progression by one nucleotide along the mRNA leader. One additional slow step was added to reflect the process of eIF1.2 dissociation. The probability distribution for completing all steps in a total time *T* – corresponding to the experimentally-observed eIF1.2 dwell-time distribution – is given by convolution of the individual probability distributions for all steps. The distribution for each step operating at rate *k* has the functional form $P(t) = ke^{-kt}$.

Taking advantage of the fact that convolution of these functions in real space is equivalent to multiplication in Laplace space[62], we computed the eIF1.2 dwell-time probability density functions by an inverse Laplace transform approach. Thus, the probability density function of an *N*-step scanning process where each scanning step occurs with a rate $k_{\text{scan}}$, and which is terminated by eIF1.2 departure at a rate of $k_{\text{eIF1.2}}$, is given by:

$$\mathcal{L}_s^{-1}\left[\frac{k_{\text{eIF1.2}}}{s + k_{\text{eIF1.2}}}\left(\frac{k_{\text{scan}}}{s + k_{\text{scan}}}\right)^N\right](t) \tag{2}$$

To reflect the independent, parallel operation of two potential start-site recognition events (i.e., the +25 CUG site and the downstream site), the computed probability density functions are composites of two weighted inverse Laplace transforms:

$$\frac{a}{a+b}\left(\mathcal{L}_s^{-1}\left[\frac{k_{\text{eIF1.2}}}{s + k_{\text{eIF1.2}}}\left(\frac{k_{\text{scan,1}}}{s + k_{\text{scan,1}}}\right)^{N_1}\right](t)\right) + \frac{b}{a+b}\left(\mathcal{L}_s^{-1}\left[\frac{k_{\text{eIF1.2}}}{s + k_{\text{eIF1.2}}}\left(\frac{k_{\text{scan,2}}}{s + k_{\text{scan,2}}}\right)^{N_2}\right](t)\right) \tag{3}$$

where *a* and *b* are the relative prevalence of commitment to the +25 and downstream sites, respectively. Subscripts 1 and 2 denote the +25 and downstream recognition processes.

### In vitro cyst counting

Confluent monolayers of HFFs on coverslips in 6-well plates were infected with either 2500 of PruΔ*ku80LUC*tdTomato-ATG8 WT and eIF1.2 F97L mutant parasites, or 10,000 of ME49Δ*ku80*, Δ*eif1.2*, or Δ*eif1.2::HA-eIF1.2* tachyzoites, in D10 medium (5% CO₂). Parasites were initially cultured in HFFs in D10 medium (5% CO₂) for 24 h before exposure to alkaline-stress medium (ambient CO₂) for 7 days. Medium was replaced daily. After 7 days of differentiation, parasites were fixed with 4% formaldehyde and subjected to immunofluorescence staining (see the immunofluorescence assays section for details). Subsequently, parasites samples were anonymized for blinded cyst counting using phase contrast microscopy conducted with a Zeiss Axio Observer Z1 inverted microscope, equipped with an EC Plan-Neofluar 20X/ 0.50 Ph 2 M27 objective. Cysts within a total of 16 image fields were counted for each sample.

### Immunofluorescence assays

2500 of PruΔ*ku80LUC*tdTomato-ATG8 WT and eIF1.2 F97L mutant parasites, or 10,000 of ME49Δ*ku80*, Δ*eif1.2*, or Δ*eif1.2::HA-eIF1.2* tachyzoites were used to infect confluent monolayers of HFFs on coverslips in 6-well plates in D10 medium (5% CO₂). At 24 h post-infection, parasites were switched to alkaline-stress media (ambient CO₂) for 7 days. Medium was replaced daily. After 7 days of differentiation, parasites were fixed for staining. $5.0 \times 10^4$ of DD-BFD1-Ty, DD-BFD1-TyΔ*eif1.2*, DD-HA-BFD2, or DD-HA-BFD2Δ*eif1.2* parasites were used to infect confluent monolayers of HFFs on coverslips in 6-well plates in D10 medium (5% CO₂). After 4 h of invasion, the medium was replaced with fresh D10 medium containing either the vehicle control (100% ethanol) or 3 µM Shield-1 (AOBIOUS; #AOB1848). These parasites were cultured for additional 4 days and then fixed for staining. Parasites were fixed in 4% formaldehyde for 10 min, permeabilized using 0.1% Triton X-100 in PBS for 10 min, blocked with 10% FBS in PBS (with 0.01% Triton X-100) for 30 min. Primary and secondary antibodies, along with other staining reagents, were diluted in wash

buffer (1% FBS, 1% normal goat serum, and 0.01% Triton X-100 in PBS). The samples were incubated with primary antibodies, either overnight at 4 °C or for 1 h at room temperature, followed by a subsequent 1 h incubation with secondary antibodies. Rabbit anti-BAG1 primary antibodies (1:1000; Carruthers Lab) and goat anti-rabbit Alexa Fluor 594 secondary antibodies (1:1000; Invitrogen; #A11012) were utilized. Additional staining reagents included biotinylated *Dolichos biflorus* agglutinin (1:400; Vector Laboratories; #B-1035) and streptavidin Alexa 350 (1:1000; Life Technologies; #S11249). After each step, 3 washes were performed using the wash buffer. Coverslips were mounted on slides with Mowiol (Calbiochem, #475904) or ProLong™ Glass Antifade Mountant (Invitrogen; # P36984). Images for quantification were captured on a Zeiss Axio Observer Z1 inverted microscope with an EC Plan-Neofluar 20X/0.50 Ph 2 M27 objective, using Zen blue edition (v2.6) software. Representative images for ME49Δ*ku80*, Δ*eif1.2*, or Δ*eif1.2::HA-eIF1.2* parasites in Fig. 4e were acquired with a Plan Apochromat 63x/1.40 oil ph3 M27 objective. All images in each experiment were captured using identical imaging parameters.

## Quantitative analysis of immunofluorescence assays
All analyses were conducted using Fiji[57] (v2.9.0/1.53t) software. Individual vacuoles were manually outlined based on phase-contrast image and saved as regions of interest (ROIs). These ROIs were then applied to BAG1, DBA or GFP (driven by the LDH2 promoter) images and their mean intensities were quantified. For each analyzed image, an uninfected region of the HFF monolayer was selected as the background. The corrected mean intensity of the BAG1, DBA or GFP signal for each vacuole was obtained by subtracting the mean intensity of the background region in the same image from the mean intensity of the respective signal. The corrected mean intensity for each condition was plotted in the graph.

## qRT-PCR
Tachyzoites from ME49Δ*ku80*, ME49Δ*ku80*Δ*eif1.2*, or ME49Δ*ku80*Δ*eif1.2::HA-eIF1.2* strains were used to infect T25 flasks with confluent HFFs in D10 medium (5% $CO_2$). RNA for day 0, corresponds to the undifferentiated stage, was collected 24 h after infection. Meanwhile, the remaining T25 flasks were switched to alkaline-stress medium (ambient $CO_2$). Medium was replaced daily to ensure alkaline pH. RNA for these samples were collected at days 1, 2, 3, and 4 of alkaline stress. RNA was isolated using TRizol™ Reagent according to the manufacture's protocol for RNA extraction. cDNA was synthesized using the SuperScript® III Frist-Strand Synthesis System for RT-PCR (Invitrogen; #18080051). Reactions were run with SsoAdvanced™ Universal SYBR® Green Supermix on CFX96 Touch Real-Time PCR Detection System (Bio-Rad) with CFX Manager (v3.1) software using the following parameters: 3 min at 95 °C, and 40 cycles of 15 s at 95 °C, followed by 30 s at 60 °C (for differentiation markers) or at 63 °C (for *eif1.2*). Primers for differentiation markers and loading control *TUB1* (Supplementary Data 3a) have been verified previously[63,64]. The amplification specificity of all primers has been verified by melt curve analysis, and by running qRT-PCR products on agarose gel.

## Polysome profiling
$2.5 \times 10^7$ ME49Δ*ku80*, ME49Δ*ku80*Δ*eif1.2* or ME49Δ*ku80*Δ*eif1.2::HA-eIF1.2* tachyzoites were used to infect a D150 plate with confluent HFFs under unstressed conditions (D10 medium, 5% $CO_2$). Four D150 plates were used for each genotype in each biological replicate. After 24 h of infection, two D150 plates remained under unstressed conditions, while the other two were exposed to a stressed environment (alkaline-stress medium, ambient $CO_2$). After 24 h of stress, the monolayers were rinsed with ice-cold intracellular buffer (25 mM HEPES-KOH pH 7.5, 142 mM KCl, 5 mM $MgCl_2$, 5 mM NaCl, and 100 µg/ml cycloheximide). Intracellular parasites were liberated from host cells by

scrapping into ice-cold intracellular buffer, passage through a 25-gauge needle, and filtering through a 3 µm Isopore™ membrane filter. The resulting parasite solution was pelleted at 1000 x g at 4 °C for 8 min using a chilled large tabletop centrifuge (Eppendorf). The pellet obtained from two D150 plates underwent three washes with intracellular buffer before being resuspended in 200 µl of ice-cold polysome lysis buffer (20 mM Tris pH 8.0, 140 mM KCl, 1.5 mM $MgCl_2$, 100 µg/ml cycloheximide, 1% (w/v) Triton X-100, 20 U/ml SUPERase•In™ RNase Inhibitor (Invitrogen, #AM2694), 1:200 dilution of Protease Inhibitor Cocktail III (Millipore; #539134), 8 U/ml TURBO™ DNase (Invitrogen; #AM2238). Following a 10-min incubation on ice, the lysates were centrifuged at 6010 x g at 4 °C for 5 min. The resulting supernatant was collected, flash-frozen in liquid $N_2$, and stored at −80 °C until further use.

The sucrose solutions comprised 20 mM Tris pH 8.0, 140 mM KCl, 5 mM $MgCl_2$, 100 µg/ml cycloheximide, 1 mM DTT, 20 U/mL SUPERase•In™ RNase Inhibitor, adjusted to 10% or 50% (w/v) sucrose concentration. Sucrose density gradients ranging from 10% to 50% (w/v) were prepared using the Gradient Master 108 (Biocomp), according to the manufacturer's instructions. The gradients were chilled for 30 min before use.

RNA concentration in the cell lysates were determined using a Nanodrop spectrophotometer. Following normalization based on their RNA concentration, equal amounts of RNA were layered onto the sucrose gradient. Samples within each biological replicate were centrifuged simultaneously for 2.5 h at 209,490 x g and 4 °C using the SW41Ti rotor in an LE-80 ultracentrifuge (Beckman). Subsequently, the samples were fractionated, and A260 absorbance measurements were taken using a Piston Gradient Fractionator (Biocomp) with FlowCell (v2.00 T) software.

## Library generation for RNA-seq and ribosome profiling
$4 \times 10^7$ ME49Δ*ku80* or ME49Δ*ku80*Δ*eif1.2* tachyzoites were used to infect a D150 plate with confluent HFFs under unstressed conditions (D10 medium, 5% $CO_2$). 2 of D150 plates were used for each genotype in each biological replicate. After 24 h of infection, one D150 plate was maintained under unstressed conditions, while the other D150 plate was subjected to a stressed environment (alkaline-stress medium, ambient $CO_2$). Parasites were harvested 48 h post-infection, following a previously published protocol[44]. The monolayers were rinsed with ice-cold intracellular buffer (25 mM HEPES-KOH pH 7.5, 142 mM KCl, 5 mM $MgCl_2$, 5 mM NaCl, and 100 µg/ml cycloheximide). Parasites were mechanically released from host cells by scrapping into 10 mL ice-cold intracellular buffer and then passage through a 25-gauge needle. The parasite solution was pelleted at 1200 x g at 4 °C for 10 min using a chilled large tabletop centrifuge (Eppendorf). The resulting pellet underwent 3 washes with intracellular buffer, after which it was resuspended in 750 µl of lysis buffer (20 mM Tris, pH 7.4, 150 mM NaCl, 5 mM $MgCl_2$, 1 mM DTT, 100 µg/ml cycloheximide, 1% (w/v) Triton X-100, 25 U/ml TURBO™ DNase. The lysate was passed through 25-gauge needle for 5 times, followed by centrifugation at 21,130 x g at 4 °C for 10 min in a tabletop chilled centrifuge (Eppendorf). 250 µl of the clarified lysate was transferred to a pre-chilled tube and mixed with 750 µl of TRIZOL LS (Sigma; #T3934). The mixture was flash-frozen in liquid $N_2$ and stored at −80 °C until RNA extraction for RNA-seq.

500 µl of the clarified lysate was transferred to a chilled tube for ribosome profiling. Fifty units of RNase I (Invitrogen) was added to the lysate. The mixture was Incubated for 45 min at room temperature with gentle mixing. Following this, 10 µl of 20 U/µl SUPERase•In™ RNase Inhibitor was added to stop nuclease digestion. The mixture was then transferred to an ultracentrifuge tube (Thermo Fisher; #45239) and underlaid with 0.90 ml of 1 M sucrose cushion (20 mM Tris, pH 7.4, 150 mM NaCl, 5 mM $MgCl_2$, 1 mM DTT, 100 µg/ml cycloheximide, 1 M sucrose, 20 U/ml SUPERase•In™ RNase

Inhibitor) by positioning the pipette tip at the very bottom of the tube and slowly dispensing the sucrose solution. Ribosomes were pelleted by centrifuging using a Sorvall™ MTX 150 Micro-Ultracentrifuge with an S110-AT fixed angle rotor at 260,000 x g at 4 °C for 4 h. The ribosomal pellet was resuspended in 700 µl of Qiazol reagent from the miRNeasy kit (Qiagen; #217004). The mixture was flash-frozen in liquid $N_2$ and stored at −80 °C until RNA extraction for Ribo-seq.

RNA was isolated using the miRNeasy kit following manufacturer's instructions. Sequencing libraries were generated as previously described in refs. 43,65. Briefly, total RNA was fragmented by alkaline hydrolysis. Total RNA and ribosome-protected footprints (RPFs) were isolated by PAGE-assisted size selection of fragments ranging from 28 to 34 nt. Pre-adenylated sequencing adapters with in-line barcodes were ligated to the repaired dephosphorylated 3' end of the RNA fragments using T4 RNA ligase 2, truncated K227Q (NEB; #M0315S). RNA and RPF samples were each pooled together after PAGE-assisted size selection. Ribosomal RNA was depleted using the eukaryotic Ribo-minus kit (Ambion; # A1083708) according to the manufacturer's protocol. Ligation of 5' linker using T4 RNA ligase 1 (NEB; #M0204S) was conducted after phosphorylating the 5' end of the pooled fragments with T4 PNK (NEB, #M0201S). Reverse transcription and PCR amplification of the library were followed by PAGE-assisted size selection were carried out as previously outlined[66]. Pooled libraries were submitted to 2 x 150 bp high-throughput sequencing using the Illumina HiSeq 4000 with bcl2Fastq (v2.17) software.

### Data analysis for RNA-seq and ribosome profiling
Sequencing adapters were removed using Cutadapt[67] (v3.7), UMIs were extracted, and residual reads that map to tRNA and rRNA were removed in silico with Bowtie2[68] (v2.4.2). The remaining reads were mapped to the *Toxoplasma* genome (ToxoDB, v56) with STAR[69] (v2.7.8a). Differentially expression and translational efficiency of annotated protein coding genes was determined with DESeq2 (v2.11.40.8) and RiboRex (v2.4.0), respectively[70,71]. The genome and annotation data are available on toxodb.org[72].

### Graphics
Figures 1a, b, 2a, 3a, 4a, 5c, h, 6a, and j and Supplementary Figure 9a were created with BioRender.com, released under a Creative Commons Attribution-NonCommercial-NoDerivs 4.0 International license (Academic License).

### Statistical analysis
The number of observations and biological replicates are described in the figure legends. For plaque (Fig. 2c, d) and flow cytometry analysis (Fig. 2 and Supplementary Figs. 2, 4), data were first test for normality using Prism (v10.2.0, GraphPad). Student's $t$-test was performed for normally distributed data, while Wilcoxon test was performed for non-normally distributed data. We used $t$-test to compare log-transformed parasite number (Fig. 2v) and gel shift data (Fig. 3b), assuming equal variance when appropriate. For the analysis of western blotting data (Figs. 2, 4, 5, 6), mice weight (Fig. 2t), plaque (Fig. 4c, d), we applied a linear mixed model when random effects were present across different replicates. We used a linear regression model in cases where no random effects were identified. These tests were performed in Rstudio[73] (v2023.06.1 + 524), with the lme4, lm and emmeans packages. For bioluminescence imaging (Fig. 2s), mice brain cysts (Fig. 4o), and immunofluorescence data (Figs. 2, 4, 6), data exhibited a wide range of outcome values. We log-transformed these data to account for the wide range of values for these outcomes. Subsequently, we analyzed these log-transformed data using either a linear mixed model or a linear regression model in Rstudio, utilizing the lme4, lm and emmeans packages. Mice survival data (Figs. 2u and 4p) were analyzed by

Mantel-Cox test using Prism. For qRT-PCR (Fig. 4n), we evaluated the difference for each day using Welch's $t$-tests combined with a Bonferroni correction. For in vitro cyst number (Figs. 2l and 4f), We ran a Poisson model in Rstudio, utilizing the lme4 package. RNA-seq and Ribosome profiling data were analyzed by DESeq2[70].

### Reporting summary
Further information on research design is available in the Nature Portfolio Reporting Summary linked to this article.

## Data availability
Unprocessed data from whole genome sequencing generated in this study have been submitted in the Sequence Read Archive (SRA) database under the following accession number: PRJNA1033315. The mutations identified in this study and the sequencing coverage are provided in the Supplementary Information/Source Data file. Unprocessed data from the RNA-seq and ribosome profiling generated in this study have been deposited in Gene Expression Omnibus (GEO) under the following accession number: GSE245775. The analyzed results of RNA-seq and ribosome profiling for this study are provided in the Supplementary Information/Source Data file. All remaining data associated with this study are provided within this manuscript, its supplementary files. Source data are provided with this paper. The database utilized was ToxoDB (https://toxodb.org/toxo/app). The dataset from Ramakrishnan et al.[45]. was sourced from ToxoDB (https://toxodb.org/toxo/app/search/transcript/GenesByRNASeqtgonME49_Ramakrishnan_enteroepithelial_stages_ebi_rnaSeq_RSRCDESeq), while the dataset from Pittman et al. was acquired directly from their manuscript[17]. Source data are provided with this paper.

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

## Acknowledgements

We thank Corey Powell (CSCAR at University of Michigan) for his help on statistical analysis; Peter Ulintz (University of Michigan) for his help on bioinformatics; Carruthers lab members: Matthew Riddell, Ari Garner, Nayanna Mercado Soto, Patrick Rimple, David Smith for their key technical support. We thank the contributors to VEuPathDB and ToxoDB for these invaluable resources to our work. We thank Biorender for providing a user-friendly tool for creating schematic illustrations. We thank ChatGPT for enhancing the clarity and correctness of our English writing. This work was supported by grants from the US National Institute of Health, including R01 AI120607 (V.B.C.), R21AI160610 (V.B.C.), F30AI169762 (P.T.), R01AI172752 (W.J.S.), R21AI167662 (W.J.S.), the Grace M. Showalter Trust (W.J.S.), R01AI158501 (S.L.), R01GM138939 (to S.O'L.), R00GM111858 (to S.O'L.), Chan Zuckerberg Initiative (J.B.Q.) and NCI P30 CA046592 (J.B.Q).

## Author contributions

F.W. and V.B.C conceived the study. F.W., M.J.H., and H.J.H. performed most of the experiments and all the data analyses. F.W. and W.D. performed polysome profiling experiment. P.T., G.K., M.H.H., and T.L.S. assisted in experiments. M.H.L. and S.L. provided plasmids, parasite strains, and information thereof. W.J.S., S.E.O., J.B.Q. supervised the study. F.W., M.J.H., H.J.H. generated the illustrations. F.W., V.B.C, H.J.H, S.E.O., and M.J.H. wrote the manuscript with input from all authors.

## Competing interests

The authors declare no competing interests.
