## [Peer Review File · Nature Communications]

Translation initiation factor eIF1.2 promotes *Toxoplasma* stage conversion by regulating levels of key differentiation factorsREVIEWER COMMENTS

Reviewer #1 (Remarks to the Author):

Wang et al

This paper describes the identification of a eIF1 homolog in *Toxoplasma* that is required for bradyzoite development. The protein was discovered serendipitously via a ENU-based autophagy screen and the authors do a commendable job of identifying the mutation that blocks differentiation and demonstrating how it likely functions during development. Overall, the work is well executed, properly controlled and interpreted, original, and significant. The only significant comment I have is that since translation initiation is still occurring, albeit at reduced levels, in the eIF1.2 knockout that providing A280 scans of polysome profiles from sucrose gradients would be useful to show in order to assess the global impact on various translation complexes during differentiation.

Other minor comments:

1. Please provide quantification of in vitro bradyzoite cyst images.
2. Please provide the images from the gel shift assays quantified in Figure 3B/
3. References to Table S2 were confusing because of the multiple worksheets (e.g. Tables S2A, S2B, etc...).
4. Please add BFD1 and BFD2 to the tables so that they are easier to find when searching the data.
5. Labels in Figure 4B need to be fixed.
6. Addition of some discussion of the various reasons why transcript abundances are impacted in the eIF1.2 knockout would be useful.
7. Scales for heatmaps are missing in Figure 5.

Reviewer #2 (Remarks to the Author):

this paper identifies eif1.2 in *T. gondii* as playing a role in bradyzoite and/or cyst formation. It uses new single molecule approaches to examine the mechanism of how this protein stabilizes translation/translation initiation and riboseq to examine the impact on overall mRNA loading onto ribosomes.

The data support the claims for the most part and are important in the field. In terms of novelty, the fact that stabilization of mRNAs promotes BZ development is known and that aspect of the paper is not necessarily novel, but the mechanism of how this might happen and all of the players are NOT known. One broad question is how specific the eif's that *T. gondii* has actually are. would certain transcripts be more likely to interact with the eif compared to others? Can this be tested reasonably using either single molecule assay (probably a big ask) or through pulldown? stated another way: is the specialness of this eif due to it's preference for certain transcripts?

general critiques/suggestions;

DBA staining should be used for all of these assays as three are examples where DBA staining and expression of genes like BAG1 are decoupled. These data are presented in some cases but not in others. I also think there is value in presenting both cyst numbers (% cyst formation above some threshold through blinded visual inspection) along with DBA intensity, as these can provide a more comprehensive view of the impact of mutations and complementation. BAG1 expression can just be only part of the story.

the phenotype in pH is very interesting as it appears that the mutants are lysing out the monolayer. is this the case? or are they just very unhappy? while disrupting cyst formation may make them not become cysts/slow down their growth they are also growing in a very unfavorable condition. For BFD2/ROCY1 we didn't see this effect (unpublished) but rather saw that the vacuoles in the mutants were just very unhappy but weren't indicative of increased growth under pH conditions. This could be informative of how comprehensive the knockout is.

In the writing of the results for ribo profiling and RNAseq it is unclear where statistically validated results for fold-changes are being discussed. While this may be in the methods it would be helpful to discuss the genes in the context of statistical significance.

Is there value in testing a real *T. gondii* transcript in the eif assay? I think it would increase the impact as it also seems to me that not every transcript gets stabilized by eif2a based on the RIBO seq data if I am reading it correctly.

Around line 110: were cysts quantified in any way in vitro using dolichos staining or some other measure besides BAG1 for the mutants? it seems from supp fig 2a that the mutants are lysing the monolayer but i think cyst formation itself should be quantified. this will help put the mutant in context of other cyst formation mutants like BFD1 and BFD2/ROCY1.

figure 4: DBA-positive cyst formation should be quantified for the knockouts and complemented strains. DBA intensity could also differ. also the "delta" symbol didn't reproduce properly (it's a square...figure 5 same thing)

Figure 3e: can you have a legend for the colors? would help even though they are indicated in figure 3c.

for in vivo cysts were there any differences in the DBA binding to the cyst wall of the mutant cysts compared to WT?

please indicate what is meant by DBA fluorescence intensity (mean across the whole vacuole?) in the figure.

Reviewer #3 (Remarks to the Author):

We mainly reviewed single-molecule analyzes of how the eIF1.2 F97L mutation affects ribosome scanning during translation initiation.

They used a very old Pacific Biosciences RS II system, which is no longer commercially available, to monitor the behavior of the translation initiation process. The author should mention where RS II is available if one want to check the correctness of the same experiment.

In the zero-mode waveguide of the RS II system, PIC and eIF1 are labeled with green and red fluorescent signals to track their activity. In particular, the duration of the green (or red) signal can measure the initial ribosome scanning and is called the dwell time (dwell time needs to be formally defined in this manuscript). The authors should mention that the current PacBio Sequell II and Revio systems do not have such capabilities and the older RS II system is the only option to perform the experiments.

Figures 3c to 3e are difficult to understand and should be completely revised according to Figure 1 in the companion paper:

Hong, H. J., Zhang, A. L., Conn, A. B., Blaha, G. & O'Leary, S. E. Single-Molecule Tracking Reveals Dynamic Regulation of Ribosomal Scanning. 2023.09.04.555162 Preprint at <https://doi.org/10.1101/2023.09.04.555162> (2023).

Figure 1 and its explanation in this preprint are well written and easy to understand.

In Figure 3f, are WT and F97L statistically significantly different in the cumulative probability of eIF1.2 residence time? No statistical analysis was performed in the manuscript.

Reviewer #4 (Remarks to the Author):

In this manuscript, Fengrong et al. provide novel insights into the regulatory mechanisms controlling the conversion of the *T. gondii* parasite from the tachyzoite form to the cyst-contained bradyzoite form, which represents the persistent stage of the parasite in its intermediate hosts. This is a fascinating and important topic lying at the interface between parasitology and immunology, as cysts are present in infected humans and are the source of complications when the immune system is weakened or suppressed.

Overall, the manuscript is well written and the data were obtained using appropriate, cutting-edge techniques, particularly the single-molecule scanning assays performed with purified eIF1.2 wt and F97L mutant proteins. The data are described in sufficient detail to support the conclusions drawn by the authors. The results obtained are interesting, featuring a well-designed set of experiments aimed at identifying and characterizing mutant parasites in eIF1.2 that are unable to convert into bradyzoites. Importantly, the eIF1.2 mutant does not impede the acute infection, indicating that the observed phenotype is not a consequence of reduced ability of the parasites to disseminate in the host organism.

As a microbiologist and biochemist, I find this work of a particular interest to a broad audience, as it may unveil new aspects of the regulatory mechanisms controlling the stage conversion process of *T. gondii*, involving proteins unique to the Apicomplexa phylum.

Conceptually, nonetheless, it is less clear why the authors chose to search for mutants with defects (or increased) in bradyzoite autophagy specifically to obtain mutants with reduced ability to convert into bradyzoites. Is autophagic flux indeed higher or lower in bradyzoites compared to tachyzoites? The current text (result section, line 78) suggests an understanding of a connection between autophagy and stage conversion. The rationale behind the autophagy process and its connection to bradyzoite differentiation needs to be explained. Please introduce what is known about autophagy in bradyzoites and what is the link with translation if any.

This conceptual concern is further illustrated by the series of mutant isolated through the genetic screen based on chemical mutagenesis and genomic sequencing. The authors isolated 8 clones or mutant strains exhibiting increased autophagic reporter expression. While the mutations are detailed in supplementary Table 1, only clone 5E4 appears to have been characterized, with no commentary in the text regarding the other clones and their identified mutations. Please indicate whether mutations in eIF1.2 were found in the other clones or in any other protein associated with autophagy functions. Such information would help support the conclusion that the eIF1.2 pathway plays a crucial role in parasite conversion.

Note that I lack the expertise to evaluate the scanning simulations presented in the extended data Figs. 5 and 6.

Other suggestions/comments.

Line 84 and Fig.1b. It is not immediately apparent to observe an elevation in tdTomato-ATG8 signal in the post-sort 3 panel when compared to the pre-sort panel. Could the authors consider incorporating additional data analysis graphics, such as Mean Fluorescence Intensity (MFI) of the selected gate for GFP and ATG8 signals? This addition could enhance the persuasiveness of the results.

Fig.2r. A main result of this study is the consequence of the F97L mutation in eIF1.2 protein on the tachyzoite to bradyzoite differentiation process. To ensure that this mutation is responsible for this phenotype in vivo, in mouse model of toxoplasmosis, it would be appreciated to demonstrate that complementing the deficient parasite with a wild-type allele of eIF1.2 reverses the phenotype of parasite burden in the brain of infected animals.

Fig.3b. The raw data for the gel shift assay is not provided. Is there any possibility of including them as supplementary data?

Line 32. Might be changed to "...defective in upregulating bradyzoite...".

Line 855. Ext. data fig.2. It could be replaced by the following text "b-m, Flow cytometry analysis...".

Line 115. Acute and chronic toxoplasmosis were reported to be assessed in CBA/J mice, but there is mention of C57BL/6 female mice in the legend of figure 2o (line 904) and the Methods section (line 538). Please provide clarification.

Line 156. I'd say more like an A to C substitution regarding the native start codon 25 nucleotides from the 5' end.

Extended Data Fig.4. Were the parasites exposed to alkaline stress in order to obtain the GFP+ signal in this assay?

Line 1020. Might be changed to "...exposed for 7 days to alkaline stress."

Line 1026. Please change to "qRT-PCR".

Line 181. It seems inappropriate to assert that eIF1.2 protein levels remain consistent throughout the differentiation stress, especially considering that the figure shown (fig.4f) indicates a significantly higher amount of eIF1.2 protein after 7 days of stress compared to day 0. Figs. 2q and 4l. Survival curve of infected mice. I might have overlooked this information, but could you please clarify the number of animals used per group in this assay?

Fig. 5d and e. I am not sure about how to interpret this figure. Is there a log2 expression scale? The bar on the right of the figure appears transparent in my version, making it unclear to me what values the blue and red bars correspond to. I am also curious why the authors opted for data from Pittman et al., which has a lower sequencing depth compared to other datasets like those from Ramakrishnan et al., available on ToxoDB.org and are a lot better to identify genes differentially regulated between tachyzoites and bradyzoites. Still along the same line, it could benefit the reader if gene names for CST4 were explicitly mentioned in the article. This is crucial due to variations across different ToxoDB releases; for instance, TGME49_261650 is listed in supplementary Table 2f, while the latest ToxoDB release includes TGME49_500108 and TGME49_500153.

Lines 264 and 267. Change to "(Fig. 6g-i)".

Line 285. Change to "(Fig. 6g-j)".

Responses to Reviewers' Comments

Manuscript ID: NCOMMS-23-55723-T

REVIEWER COMMENTS

We sincerely thank all the reviewers for their valuable feedback, constructive suggestions, and insightful comments on our manuscript. We have carefully considered each point raised and have made revisions accordingly. Below, we provide point-to-point responses to address the reviewers' comments.

Reviewer #1 (Remarks to the Author):

Wang et al

This paper describes the identification of a eIF1 homolog in *Toxoplasma* that is required for bradyzoite development. The protein was discovered serendipitously via a ENU-based autophagy screen and the authors do a commendable job of identifying the mutation that blocks differentiation and demonstrating how it likely functions during development. Overall, the work is well executed, properly controlled and interpreted, original, and significant. The only significant comment I have is that it since translation initiation is still occurring, albeit at reduced levels, in the eIF1.2 knockout that providing A280 scans of polysome profiles from sucrose gradients would be useful to show in order to assess the global impact on various translation complexes during differentiation.

Many thanks for the reviewer's positive feedback. We greatly appreciate your suggestion, and agree that polysome profiling would be useful to provide a global view of translation complexes. Accordingly, we performed polysome profiling experiments using both unstressed and 1 day alkaline stressed WT (ME49 $\Delta ku80$), $\Delta eif1.2$, or $\Delta eif1.2::HA-eIF1.2$ parasites. Our results revealed well resolved polysome peaks in unstressed WT, $\Delta eif1.2$, or $\Delta eif1.2::HA-eIF1.2$ parasites, indicative of active protein translation. After 1 day of alkaline stress, all genotypes showed diminished polysome peaks, suggesting a general downregulation of translation. The corresponding data has been incorporated into Extended Data Fig. 8a,b.

Other minor comments:

1. Please provide quantification of *in vitro* bradyzoite cyst images.

We added quantification of *in vitro* bradyzoite cyst images for the Pru strain background in Fig. 2I, and for the ME49 strain background in Fig. 4f.

2. Please provide the images from the gel shift assays quantified in Figure 3B/

We have included images of gel shift assays in Extended Data Fig. 5.

3. References to Table S2 were confusing because of the multiple worksheets (e.g. Tables S2A, S2B, etc...).

Thank you for pointing this out. To address the confusion, we have now indicated in the text to refer to the specific worksheet in Table S2.

4. Please add BFD1 and BFD2 to the tables so that they are easier to find when searching the data.

We have revised the tables to renamed TGME49_200385 (Myb family DNA-binding domain-containing protein) to TGME49_200385 (BFD1) and TGME49_311100 (zinc finger (CCCH type) motif-containing protein) to TGME49_311100 (BFD2).

5. Labels in Figure 4B need to be fixed.

Thank you for the suggestion. We apologize for any confusion regarding the labels requiring adjustment. We suspect that you are referring to the concern raised by reviewer #4 regarding the incorrect reproduction of the “delta” symbol as a square? We have verified in the version we uploaded to the journal through our account, and we observe the correct “delta” symbol, not a square. We are unsure why the version provided to reviewers displays a square instead. We suspect that perhaps the “delta” symbol we initially used wasn’t recognized properly by the journal’s system. To address this, we have reinserted the “delta” symbol in Figs. 4-6 using a different method. We hope this resolves the issue and the “delta” symbol now appears correctly on your end.

6. Addition of some discussion of the various reasons why transcript abundances are impacted in the eIF1.2 knockout would be useful.

We have added to the discussion the following text outlining possible reasons for the observed impact on transcript abundances resulting from the absence of eIF1.2.

“The observed influence of eIF1.2 deficiency on transcript abundances underscores the intricate interplay between translation initiation and gene expression. We propose several mechanisms through which eIF1.2 impacts mRNA levels: first, by affecting the translation of transcription factors (such as BFD1) or regulatory proteins, thus influencing the transcriptional activity of specific genes; second, by potentially triggering cellular stress responses or signaling pathways that modulate transcriptional regulation; and finally, by potentially altering the association of RNA-binding proteins with mRNA transcripts, thereby affecting mRNA stability and abundance. Our findings suggest that eIF1.2 might play a multifaceted regulatory role in orchestrating gene expression dynamics in *T. gondii*.”

7. Scales for heatmaps are missing in Figure 5.

We apologize for the missing scales on the heatmaps. The heatmaps were generated in Prism as images, inserted into a PowerPoint file as figures, and then included in our Word document containing the manuscript. The scale is present in our Word file. It’s possible that the scale was lost during the uploading process. We will ensure to double-check the Word file once it is uploaded to ensure the scales are properly included, and we may also reach out to technical support for assistance in resolving this issue.

Reviewer #2 (Remarks to the Author):

this paper identifies eif1.2 in *T. gondii* as playing a role in bradyzoite and/or cyst formation. It uses new single molecule approaches to examine the mechanism of how this protein stabilizes translation/translation initiation and riboseq to examine the impact on overall mRNA loading onto ribosomes.

We deeply value the constructive suggestions provided by the reviewer. Your insightful input for improvement is highly appreciated.

The data support the claims for the most part and are important in the field. In terms of novelty, the fact that stabilization of mRNAs promotes BZ development is known and that aspect of the paper is not necessarily novel, but the mechanism of how this might happen and all of the players are NOT known. One broad question is how specific the eif's that *T. gondii* has actually are. would certain transcripts be more likely to interact with the eif compared to others? Can this be tested reasonably using either single molecule assay (probably a big ask) or through pulldown? stated another way: is the specialness of this eif due to it's preference for certain transcripts?

Thank you very much for your questions and suggestions. We attempted CLIP-seq with HA-tagged eIF1.2. Since eIF1.2 associates with ribosomes, we anticipated significant ribosomal RNA pulled down with eIF1.2. Despite performing rRNA depletion to isolate genuine targets, obtaining sufficient mappable reads posed challenges. In hindsight, it seems unlikely that such experiments would be successful, as eIF1.2 does not directly bind to mRNA. If a hypothetical "protein X" were to interact directly with both mRNA and eIF1.2, the mRNA•X•eIF1 complex might not withstand RNA immunoprecipitation sequencing (RIP-seq) conditions. Therefore, conducting the differential expression experiment appears to be the most feasible approach at this stage. We appreciate your valuable input and will continue to explore methods to elucidate specific transcripts for eIF1.2.

general critiques/suggestions;

DBA staining should be used for all of these assays as three are examples where DBA staining and expression of genes like BAG1 are decoupled. These data are presented in some cases but not in others. I also think there is value in presenting both cyst numbers (% cyst formation above some threshold through blinded visual inspection) along with DBA intensity, as these can provide a more comprehensive view of the impact of mutations and complementation. BAG1 expression can just be be only part of the story.

We have incorporated the quantification of *in vitro* cyst numbers (through visual enumeration of blinded samples) and the mean fluorescence intensity for DBA and GFP expression driven by LDH2 promoter for the Pru strain into Fig. 2l-o. Similarly, for the ME49 strain, we have included the quantification of *in vitro* cyst numbers (through visual enumeration of blinded samples) and the mean fluorescence intensity for DBA and BAG1 in Fig. 4f-h.

the phenotype in pH is very interesting as it appears that the mutants are lysing out the monolayer. is this the case? or are they just very unhappy? while disrupting cyst formation may make them not become cysts/slow down their growth they are also growing in a very

unfavorable condition. For BFD2/ROCY1 we didn't see this effect (unpublished) but rather saw that the vacuoles in the mutants were just very unhappy but weren't indicative of increased growth under pH conditions. This could be informative of how comprehensive the knockout is. Thank you for your question. Both eIF1.2 F97L mutants in the Pru background, and $\Delta eif1.2$ mutants in the ME49 background appeared to thrive under alkaline stress conditions. By phase contrast microscopy, we observed their robust growth in alkaline media, seemingly unaffected by the stress. We have tried to infect host cells with eIF1.2 F97L parasites at varying MOIs and exposed them to alkaline stress for up to 7 days (unpublished). At very low MOI, they did not form cysts and instead behaved like tachyzoites, lysing small regions of host cells, and then invading new ones. At higher MOIs, they lysed out the entire host monolayer. At the same MOIs, WT parasites develop robust cysts. These observations suggest to us that they can thrive under alkaline stress, like WT tachyzoites growing in regular media without stress.

Thank you for sharing the phenotype of the BFD2/ROCK1 mutants with us. We similarly noticed that alkaline stressed BFD1 KO:DD-BFD1-Ty parasites appeared unhealthy and reside in deformed vacuoles. We speculate that the difference between eIF1.2 mutants and BFD1 mutants, as well as BFD2/ROCK1 mutants, may be due to eIF1.2's position upstream of BFD1 and BFD2/ROCK1. The absence of eIF1.2 may affect other genes in addition to BFD1 and BFD2/ROCK1, which could contribute to dealing with alkaline stress and facilitate the growth of eIF1.2 mutants under stressed conditions.

In the writing of the results for ribo profiling and RNAseq it is unclear where statistically validated results for fold-changes are being discussed. While this may be in the methods it would be helpful to discuss the genes in the context of statistical significance.

fold change and P value cutoffs or P_{adj} values have been included for each discussed genes in the text on Pages 6.7.

Is there value in testing a real *T. gondii* transcript in the eif assay? I think it would increase the impact as it also seems to me that not every transcript gets stabilized by eif2a based on the RIBO seq data if I am reading it correctly.

Many thanks for your suggestion. Because eIF1.2 does not directly bind to RNAs, we currently lack an effective method to identify specific *T. gondii* transcripts associated with eIF1.2. Therefore, we are unable to perform such experiment at this time.

Around line 110: were cysts quantified in any way in vitro using dolichos staining or some other measure besides BAG1 for the mutants? it seems from supp fig 2a that the mutants are lysing the monolayer but i think cyst formation itself should be quantified. this will help put the mutant in context of other cyst formation mutants like BFD1 and BFD2/ROCY1. We have incorporated quantification of cyst numbers in Fig. 2I.

figure 4: DBA-positive cyst formation should be quantified for the knockouts and complemented strains. DBA intensity could also differ. also the "delta" symbol didn't reproduce properly (it's a square...figure 5 same thing)

We have added quantification for DBA mean fluorescence intensity within individual vacuoles for ME49 WT, $\Delta eif1.2$, or $\Delta eif1.2::HA-eIF1.2$ parasites. Thank you for bringing up the issue with “delta” symbol. We have verified in the version we uploaded to the journal through our account, and we observe the correct “delta” symbol, not a square. We are unsure why the version provided to reviewers displays a square instead. We suspect that perhaps the “delta” symbol we initially used wasn’t recognized properly by the journal’s system. To address this, we have reinserted the “delta” symbol in Figs. 4-6 using a different method. We hope this resolves the issue and the “delta” symbol now appears correctly on your end.

Figure 3e: can you have a legend for the colors? would help even though they are indicated in figure 3c.

Given the word limit constraint for figure legend, it is challenging to add extra text in the legend for Fig. 3. Instead, we addressed this by updating Fig. 3e (now Fig. 3f) with colored names corresponding to the traces and colors used in Fig. 3c (now Fig. 3e).

for in vivo cysts were there any differences in the DBA binding to the cyst wall of the mutant cysts compared to WT?

Pru $\Delta ku80$ bradyzoites (WT and eIF1.2 F97L mutant) are intrinsically difficult to quantify accurately via phase contrast microscopy because of their low numbers and small size. While attempting to quantify them by assessing the GFP signal (driven by LDH2 promoter), we observed that some structures appearing green were, in fact, autofluorescent debris rather than cysts. Consequently, we chose to utilize qRT-PCR to quantify parasite numbers in the mouse brain instead of relying on cyst counting. We have in the past performed DBA staining of ME49 *ex vivo* cysts; however, we were somewhat leery of the results upon noting that some cysts were not DBA positive. ME49 cysts tend to be large, abundant, and quite easily recognizable by phase contrast microscopy by a trained eye.

please indicate what is meant by DBA fluorescence intensity (mean across the whole vacuole?) in the figure.

Thank you for your suggestion. We have updated the y-axis title to “DBA mean fluorescence intensity” and clarified in the figure legend that we quantified mean fluorescence intensity for DBA within individual vacuoles in Figs. 2,4,6.

Reviewer #3 (Remarks to the Author):

We mainly reviewed single-molecule analyzes of how the eIF1.2 F97L mutation affects ribosome scanning during translation initiation.

We sincerely appreciate the reviewer’s expertise in single-molecule experiments, which has significantly contributed to enhancing the quality and robustness of our manuscript.

They used a very old Pacific Biosciences RS II system, which is no longer commercially available, to monitor the behavior of the translation initiation process. The author should mention where RS II is available if one want to check the correctness of the same experiment.

We have updated the methods section to include that RS II is available at the University of California, Riverside.

In the zero-mode waveguide of the RS II system, PIC and eIF1 are labeled with green and red fluorescent signals to track their activity. In particular, the duration of the green (or red) signal can measure the initial ribosome scanning and is called the dwell time (dwell time needs to be formally defined in this manuscript). The authors should mention that the current PacBio Sequell II and Revio systems do not have such capabilities and the older RS II system is the only option to perform the experiments.

We have incorporated the definition of dwell time into our main text on page 4, as well as included a description of dwell time in the figure legend for Fig. 3. Additionally, we have emphasized in our methods section that the current PacBio Sequell II and Revio systems lack the capability for these experiments, highlighting the older RS II system is the only option available for conducting them.

Figures 3c to 3e are difficult to understand and should be completely revised according to Figure 1 in the companion paper:

Hong, H. J., Zhang, A. L., Conn, A. B., Blaha, G. & O'Leary, S. E. Single-Molecule Tracking Reveals Dynamic Regulation of Ribosomal Scanning. 2023.09.04.555162 Preprint at <https://doi.org/10.1101/2023.09.04.555162> (2023).

Figure 1 and its explanation in this preprint are well written and easy to understand.

We have revised Fig.3 and its accompanying figure legend in alignment with the Fig. 1 from the companion paper.

In Figure 3f, are WT and F97L statistically significantly different in the cumulative probability of eIF1.2 residence time? No statistical analysis was performed in the manuscript.

Yes, there is a statistically significant difference between in the cumulative probability of eIF1.2 residence time. We have included *P* values in the table of Fig. 3g and specified in the figure legend that the paired WT/F97L distributions within replicate experiments were statistically different (*P* = 0.0126 for replicate #1, *P* = 0.0005 for replicate #2; Wilcoxon rank-sum test).

Reviewer #4 (Remarks to the Author):

In this manuscript, Fengrong et al. provide novel insights into the regulatory mechanisms controlling the conversion of the *T. gondii* parasite from the tachyzoite form to the cyst-contained bradyzoite form, which represents the persistent stage of the parasite in its intermediate hosts. This is a fascinating and important topic lying at the interface between parasitology and immunology, as cysts are present in infected humans and are the source of complications when the immune system is weakened or suppressed.

Overall, the manuscript is well written and the data were obtained using appropriate, cutting-edge techniques, particularly the single-molecule scanning assays performed with purified

eIF1.2 wt and F97L mutant proteins. The data are described in sufficient detail to support the conclusions drawn by the authors. The results obtained are interesting, featuring a well-designed set of experiments aimed at identifying and characterizing mutant parasites in eIF1.2 that are unable to convert into bradyzoites. Importantly, the eIF1.2 mutant does not impede the acute infection, indicating that the observed phenotype is not a consequence of reduced ability of the parasites to disseminate in the host organism.

As a microbiologist and biochemist, I find this work of a particular interest to a broad audience, as it may unveil new aspects of the regulatory mechanisms controlling the stage conversion process of *T. gondii*, involving proteins unique to the Apicomplexa phylum.

We greatly appreciate your positive and encouraging comments!

Conceptually, nonetheless, it is less clear why the authors chose to search for mutants with defects (or increased) in bradyzoite autophagy specifically to obtain mutants with reduced ability to convert into bradyzoites.

We apologize for not making this clear in our text. Based on our lab's prior research highlighting the importance of autophagy for bradyzoite viability (Di Cristina *et al.*, 2017) and identifying CPL and ATG9 as genes linked to bradyzoite autophagy (Di Cristina *et al.*, 2017; Smith *et al.*, 2021), we employed chemical mutagenesis combined with flow sorting to uncover additional genes involved in bradyzoite autophagy. Unexpectedly, during the process, we fortuitously identified a gene (eIF1.2) implicated in differentiation. This discovery has been both unexpected and exhilarating for us. Upon subjecting the 8 individual mutants to alkaline stress for 7 days to confirm their enhanced tdTomato-ATG8 signals in the bradyzoite stage, we observed that 5-E4 (harboring multiple mutations) displayed poor cyst formation under phase contrast microscopy. This distinctive phenotype is readily apparent even upon cursory inspection under phase contrast microscopy.

Is autophagic flux indeed higher or lower in bradyzoites compared to tachyzoites?

We have not investigated this aspect, but it presents an intriguing avenue for future exploration.

The current text (result section, line 78) suggests an understanding of a connection between autophagy and stage conversion. The rationale behind the autophagy process and its connection to bradyzoite differentiation needs to be explained. Please introduce what is known about autophagy in bradyzoites and what is the link with translation if any.

We are sorry for the confusion. Currently, we do not know if there is a connection between autophagy and stage conversion. None of the other 7 autophagy mutant clones showed an apparent differentiation defect upon initial inspection under phase contrast microscopy. We have revised this section to ensure that we do not imply a link between autophagy with differentiation.

This conceptual concern is further illustrated by the series of mutant isolated through the genetic screen based on chemical mutagenesis and genomic sequencing. The authors isolated 8 clones or mutant strains exhibiting increased autophagic reporter expression. While the mutations are detailed in supplementary Table 1, only clone 5E4 appears to have been

characterized, with no commentary in the text regarding the other clones and their identified mutations. Please indicate whether mutations in eIF1.2 were found in the other clones or in any other protein associated with autophagy functions. Such information would help support the conclusion that the eIF1.2 pathway plays a crucial role in parasite conversion.

Thank you for your suggestion. eIF1.2 was exclusively identified in clone 5E4. We have emphasized this by including the statement “Notably, there were no mutations shared among these 8 independent clones” on page 3.

Note that I lack the expertise to evaluate the scanning simulations presented in the extended data Figs. 5 and 6.

Other suggestions/comments.

Line 84 and Fig.1b. It is not immediately apparent to observe an elevation in tdTomato-ATG8 signal in the post-sort 3 panel when compared to the pre-sort panel. Could the authors consider incorporating additional data analysis graphics, such as Mean Fluorescence Intensity (MFI) of the selected gate for GFP and ATG8 signals? This addition could enhance the persuasiveness of the results.

We have added additional flow cytometry pseudocolored plots for the GFP and tdTomato-ATG8 channels. As different voltage settings were used for GFP and tdTomato-ATG8 for before and after the three rounds of sorting, direct comparison of absolute median fluorescence intensity (MFI) values is not feasible. Therefore, we quantified MFI ratio (mut pop'n/WT) for GFP and tdTomato-ATG8 signals before and after the three rounds of sorting, as depicted in Fig. 1.

Fig.2r. A main result of this study is the consequence of the F97L mutation in eIF1.2 protein on the tachyzoite to bradyzoite differentiation process. To ensure that this mutation is responsible for this phenotype in vivo, in mouse model of toxoplasmosis, it would be appreciated to demonstrate that complementing the deficient parasite with a wild-type allele of eIF1.2 reverses the phenotype of parasite burden in the brain of infected animals.

We greatly appreciate your suggestion regarding the potential demonstration of phenotype reversal in the mouse model by complementing eIF1.2 F97L parasites with a wild-type allele of eIF1.2. Regrettably, we were unable to pursue this approach due to inability to introduce the wild-type allele into the native locus, as eIF1.2 with the F97L mutation is present at that locus. We were concerned that introducing the wild-type allele elsewhere might impact gene expression and not fully complement the phenotype. We believe our current results provide robust support for the conclusion that the F97L mutation is responsible for the observed phenotype. Firstly, we identified the F97L mutation in the 5-E4 mutant, which harbors multiple mutations, through chemical mutagenesis. This mutant exhibited clear differentiation defects. Secondly, we utilized the CRISPR-Cas9 method to specifically introduce the F97L mutation into parasites. In both cases, parasites carrying eIF1.2 F97L mutation displayed differentiation defects. The consistent phenotype observed in mutants generated through two distinct methods strongly supports the conclusion that F97L mutation is responsible for the differentiation defects.

Fig.3b. The raw data for the gel shift assay is not provided. Is there any possibility of including them as supplementary data?

We have included images of gel shift assays in Extended Data Fig. 5.

Line 32. Might be changed to "...defective in upregulating bradyzoite...".

We have made the correction.

Line 855. Ext. data fig.2. It could be replaced by the following text "b-m, Flow cytometry analysis...".

Yes, we have made the correction according to your suggestion.

Line 115. Acute and chronic toxoplasmosis were reported to be assessed in CBA/J mice, but there is mention of C57BL/6 female mice in the legend of figure 2o (line 904) and the Methods section (line 538). Please provide clarification.

Yes, it should be C57BL/6 mice. We have made the correction.

Line 156. I'd say more like an A to C substitution regarding the native start codon 25 nucleotides from the 5' end.

Yes, you are correct. We have made the correction.

Extended Data Fig.4. Were the parasites exposed to alkaline stress in order to obtain the GFP+ signal in this assay?

Yes, the parasites were indeed exposed to alkaline stress for 7 days to obtain the GFP+ signal in this assay. We have incorporated this information into the figure legend for Extended Data Fig. 4.

Line 1020. Might be changed to "...exposed for 7 days to alkaline stress."

We have updated the text accordingly to your recommendation.

Line 1026. Please change to "qRT-PCR".

We have updated the text accordingly to your recommendation.

Line 181. It seems inappropriate to assert that eIF1.2 protein levels remain consistent throughout the differentiation stress, especially considering that the figure shown (fig.4f) indicates a significantly higher amount of eIF1.2 protein after 7 days of stress compared to day 0.

Thank you for your suggestion. You are correct in highlighting the potential limitations of drawing conclusions about eIF1.2 protein levels based on the initial 3 biological replicates, which exhibited considerable variability. We acknowledge this limitation and have taken steps to address it. Specifically, we conducted additional 5 biological replicates, resulting in a total of 8 replicates for analysis. With this expanded data, we observed no statistical significance in eIF1.2 levels between the unstressed condition (Day 0) and Day 1 after stress. However, we did observe a statistically significant decrease for eIF1.2 levels at Day 7 after stress compared to Day 1 after stress. To reflect these findings, we have made updated the representative image

for the western blot (Fig. 4i) and included the quantification data for all 8 biological replicates in Fig. 4j.

Figs. 2q and 4l. Survival curve of infected mice. I might have overlooked this information, but could you please clarify the number of animals used per group in this assay?

You are correct that the number of mice used in generating the survival curve was omitted. We apologize for this oversight. Due to the word limit (up to 350 words) for figure legends, we were unable to include this information in the legend for Fig. 2. Instead, we have provided this information for both Figs. 2 and 4 in the methods section (refer to pages 14 and 15). For Fig. 2, total number of mice used for infection to generate the survival curve is as follows: WT (n=12), eIF1.2 F97L (n=12). For Fig. 4, total number of mice used for infection to generate the survival curve is as follows: WT (n=20), $\Delta eif1.2$ (n=20), and $\Delta eif1.2::HA-eIF1.2$ (n=20).

Fig. 5d and e. I am not sure about how to interpret this figure. Is there a log₂ expression scale? The bar on the right of the figure appears transparent in my version, making it unclear to me what values the blue and red bars correspond to.

Yes, you are correct, and we apologize for the missing scales on the heatmaps. The heatmaps were generated in Prism as images, inserted into a PowerPoint file as figures, and then included in our Word document containing the manuscript. The scale is present in our Word file. It's possible that the scale was lost during the uploading process. We will ensure to double-check the Word file once it is uploaded to ensure the scales are properly included, and we may also reach out to technical support for assistance in resolving this issue.

I am also curious why the authors opted for data from Pittman et al., which has a lower sequencing depth compared to other datasets like those from Ramakrishnan et al., available on ToxoDB.org and are a lot better to identify genes differentially regulated between tachyzoites and bradyzoites.

We chose to cross-compare with Pittman *et al.* since it provides dataset for *in vivo* infections, spanning from tachyzoites (acute infection, 10 days post-infection) to bradyzoites stage (chronic infection, 28-post infection). Our aim was to determine whether our findings mirrored significant gene changes observed in *T. gondii* during mouse infections. To our understanding, Ramakrishnan *et al.*, provides data from *in vitro* tachyzoites, and tissue cysts (50 days post-infection). Thank you for pointing out that dataset provided by Ramakrishnan *et al.* has higher sequencing depth, making it better suited for identifying differentially regulated genes between tachyzoites and bradyzoites. We have included a comparison between our dataset and the dataset for *in vitro* tachyzoites and tissue cysts from Ramakrishnan *et al.* in Extended Data Fig. 9. We observed a similar trend to when we compare our dataset with the dataset from Pittman *et al.*

Still along the same line, it could benefit the reader if gene names for CST4 were explicitly mentioned in the article. This is crucial due to variations across different ToxoDB releases; for instance, TGME49_261650 is listed in supplementary Table 2f, while the latest ToxoDB release includes TGME49_500108 and TGME49_500153.

Thank you for the valuable suggestion. We have updated supplementary Table 2f to include the gene name, CST4, for TGME49_261650. Additionally, we have included gene number TGME49_261650 in reference to CST4 in our text on page 7.

Lines 264 and 267. Change to “(Fig. 6g-i)”.
We have corrected this.

Line 285. Change to “(Fig. 6g-j)”.
Thank you for your suggestion. We apologize for the mislabeling. We meant to refer to the schematic in Fig. 6j. We have now corrected the reference from (Fig. 6i) to (Fig. 6j) accordingly.

REVIEWERS' COMMENTS

Reviewer #1 (Remarks to the Author):

The authors have addressed all my concerns. This is a wonderful paper. Congrats to all.

Reviewer #2 (Remarks to the Author):

Wonderfully thorough response to my comments. thank you. An exciting piece of work as we move further up the bradyzoite initiation pathway!! Jon

Reviewer #3 (Remarks to the Author):

I checked the revised text and figures. I confirmed that the author answered my questions satisfactorily.

Reviewer #4 (Remarks to the Author):

Overall, I agree with most of the co-reviewers' remarks. In the revised manuscript, the authors have adequately addressed the main queries and concerns raised by the reviewers. I believe the changes improve the quality of the manuscript, and I do not have any further comments. Finally, every mutagenesis followed by the selection of parasites under alkaline stress conditions would likely lead to the selection of mutants in eIF1.2, as they should confer fitness advantages specifically in these settings compared to the parental strain that differentiates into slow-growing bradyzoites.

Responses to Reviewers' Comments

Manuscript ID: NCOMMS-23-55723-B

REVIEWER COMMENTS

We wish to express our profound gratitude to all the reviewers for generously dedicating their time and expertise to evaluate our manuscript. Their insightful feedback and constructive suggestions have been instrumental in enhancing the quality of our work, and we are truly grateful for their contributions.

Reviewer #1 (Remarks to the Author):

The authors have addressed all my concerns. This is a wonderful paper. Congrats to all.

Reviewer #2 (Remarks to the Author):

Wonderfully thorough response to my comments. thank you. An exciting piece of work as we move further up the bradyzoite initiation pathway!! Jon

Reviewer #3 (Remarks to the Author):

I checked the revised text and figures. I confirmed that the author answered my questions satisfactorily.

Reviewer #4 (Remarks to the Author):

Overall, I agree with most of the co-reviewers' remarks. In the revised manuscript, the authors have adequately addressed the main queries and concerns raised by the reviewers. I believe the changes improve the quality of the manuscript, and I do not have any further comments. Finally, every mutagenesis followed by the selection of parasites under alkaline stress conditions would likely lead to the selection of mutants in eIF1.2, as they should confer fitness advantages specifically in these settings compared to the parental strain that differentiates into slow-growing bradyzoites.

Thank you very much for your insightful perspective. We wholeheartedly concur with the notion that the ability of eIF1.2 mutants to thrive akin to tachyzoites under alkaline stress conditions provides them with a fitness advantage over the parental strain, which differentiates into slow-growing bradyzoites. And this might be one of the reasons why we uncovered eIF1.2 mutants in our screen process. We have integrated this prospect into paragraph 1 of the discussion section as follows: "This finding is likely attributed to the fitness advantage acquired by eIF1.2 mutants,

which grow akin to tachyzoites under alkaline stress conditions, compared to the parental strain that differentiates into slow-growing bradyzoites under the same conditions.”